

# Headwater sediment dynamics in debris flow catchment: implication of debris supply using high resolution topographic surveys

A. Loye[1,a], M. Jaboyedoff[1], J. I. Theule[2], and F. Liébault[2]

[1]Institute of Geomatics and Risk Analysis, University of Lausanne, Lausanne, Switzerland
[2]Unité de Recherche ETNA, IRSTEA Grenoble, Saint-Martin-d'Hères, France
[a]now at: GEOTEST SA, Le Mont-sur-Lausanne, Switzerland

Received: 1 December 2015 – Accepted: 2 December 2015 – Published: 14 January 2016

Correspondence to: A. Loye (alexandre.loye@geotest.ch)

Published by Copernicus Publications on behalf of the European Geosciences Union.

---

Sidebar:

**ESURFD**

doi:10.5194/esurf-2015-48

## Abstract

Debris flows have been recognized to be linked to amounts of material temporary stored in torrent channels. Consequently, sediment production, debris supply and storage changes from low-order channels of the Manival catchment (French Alps) were surveyed periodically during 16 months using terrestrial laser scanning (TLS) to study the coupling between sediment dynamics and torrent responses in terms of debris flow events, which happened twice during the monitoring period. Sediment transfer in the main torrent was monitored with cross-section surveys. Sediment budgets were generated seasonally using sequential TLS data differencing and morphological extrapolations. Debris production depends strongly on rockfall occurring during winter–early spring season, following power law distribution for volumes of rockfall events above $0.1\,\mathrm{m}^3$, while hillslope sediment reworking dominates debris recharge from spring to autumn. Both debris flows originate in channels exclusively, but their occurrence is linked to recharge from previous debris pulses coming from the hillside and from bedload transport. Headwater debris sources display an equivocal behaviour in sediment transfer: despite of rainstorms inducing debris flows in torrent, low geomorphic activity occurred in production zone. Still, a general reactivation of sediment transport in headwater channels was observed in autumn without new debris supply, suggesting no exhaustion of debris storages. The seasonal cycle of sediment yield seems therefore to depend not only on debris supply and runoff (flow capacity), but also on geomorphic conditions that destabilize remnant debris stocks. This study shows that a monitoring of torrent in-channel storage changes coupled to debris supply can readily improve knowledge on recharge threshold leading to debris flow, so their prediction.

## 1 Introduction

In steep mountain catchments, rainfall intensity and duration (incl. snowmelt) are insufficient to predict debris flow, despite that conditions of initiation for runoff-generated

# ESURFD

doi:10.5194/esurf-2015-48

**Headwater sediment dynamics in debris flow catchment**

A. Loye et al.

debris flow require a significant water inflow (Van Dine, 1985; Decaulne and Saemundsson, 2007; Guzzetti, 2008). In many cases, the main reason arises from the fact that the amount of debris that can be entrained in a channel reach is often more significant than mechanisms of initiation (Hungr, 2011; Theule et al., 2015). The frequency and magnitude of debris flow have been recognized to be linked to the amount of material temporary stored in channel reaches (Van Steijn et al., 1996; Cannon et al., 2003; Hungr et al., 2005), such that hillside sediment delivery recharging those channels represents a key factor for the occurrence of debris flow (e.g. Benda and Dunne, 1997; Bovis and Jakob, 1999; Berti et al., 2000). This implies efficient hillslope–channel coupling (Hooke, 2003; Schlunegger et al., 2009; Johnson and Warburton, 2010). The rate of sediment supply needs therefore to be considered for predicting debris flow hazard (Rickenmann, 1999; Jakob et al., 2005). The difficulty results however in quantifying sediment process activity from hillslopes and in-channel debris storage (Peiry, 1990; Zimmermann et al., 1997).

Recording of overall sediment production and transfer rate has increasingly relied upon multi-temporal digital stereophotogrammetry (Coe et al., 1993; Chandler and Brunsden, 1995; Veyrat-Chavillon and Memier, 2006) and elevation difference from High Resolution Digital Elevation Models (HRDEM) (Smith et al., 2000; Wu and Cheng, 2005; Roering et al., 2009; Theule et al., 2012). In terrain dominated by slopes of high-steepness, traditional aerial derived DEMs remain typically inappropriate to study geomorphic processes. Limitations concern not only the too low rendering of small topographic changes (Perroy et al., 2010), this can be technically improved, but also the poor surface representation of steep terrain with small curvature radii and the data gaps in vertically oriented and overhanging topography. Even on gentler slopes, the sharp break of slopes, encountered in erosion scars for instance, was demonstrated to be insufficiently modelled by airborne HRDEM, leading to erroneous volume estimations (Bremer and Sass, 2011). This represents a serious drawback in budgeting steep terrain, where sediment activity comes mostly from rock walls and rugged gullies. Because of these issues, many hill- and rock slope process studies have been

ESURFD

doi:10.5194/esurf-2015-48

**Headwater sediment dynamics in debris flow catchment**

A. Loye et al.

Title Page

Abstract | Introduction

Conclusions | References

Tables | Figures

◁ | ▷|

◁ | ▷

## ESURFD

doi:10.5194/esurf-2015-48

**Headwater sediment dynamics in debris flow catchment**

A. Loye et al.

investigated with terrestrial laser scanner (TLS) (Jaboyedoff et al., 2012). The recent development of long range TLS devices provides an effective mean of acquiring high resolution topographic information that adequately reflect the morphology of steep bedrock-dominated areas. The practical disadvantage in data acquisition related to
ground survey can be compensated by flexibility in transport, ensuring a full coverage with minimum shadow zones.

This paper presents a quantitative study of sediment recharge and channel response leading to debris flow, using 3-D digital terrain models provided by TLS. This is illustrated on the Manival (French Alps), a torrent that experiments runoff-generated debris flow almost every year (Péteuil et al., 2008). The entire hillslope processes and sediment dynamics from tributary channels to the torrent was surveyed periodically over 16 months. The spatio-temporal variability of debris production and subsequent transport and storage of sediment are analysed on a seasonal time scale, in order to discuss the debris supply dynamics and the implications in debris flow initiation. This
study also complements the investigation about what controls debris flow erosion and bedload transport done in the Manival's torrent (Theule et al., 2015).

## 2 Study site

### 2.1 General setting

The 3.9 km$^2$ Manival catchment located at the edge of the Chartreuse massif (France) (Fig. 1) displays a rugged, 1200 m relief watershed, resulting from a deep headward entrenchment (Gidon, 1991). The topography consists of a steep-sided colluvium-filled valley made of series of rock walls and slopes mantled with scree deposits. The lithology ranges in age from late Jurassic to early Cretaceous (Fig. 2) (Charollais et al., 1986). In the heart of the basin, thick sequences of calcareous marl interbedded with layers of marl predominate. Towards the ridge, the bedrock evolves progressively from more stratified to massive limestone. The valley sides correspond to the fold limbs of

**ESURFD**

doi:10.5194/esurf-2015-48

**Headwater sediment dynamics in debris flow catchment**

A. Loye et al.

an anticline, where secondary folding and minor faults induce local variations in structure (Gidon, 1991). This tectonic setting and the varying stratigraphic competency have strongly influenced the topographic development, providing a dynamic geomorphic environment supported by an important runoff as a response to heavy rainstorms that
occur regularly.

## 2.2 Characteristic of the headwater sediment dynamic

Contemporary geomorphic activity contributing to recharge the torrent with debris concentrates exclusively in the headwater, where any remnant glacial deposits are found (Gruffaz, 1997). In the upper catchment, a large old rock deposits flooring the hills-
lope side west (Fig. 3) have dramatically influenced the bottom topography, and thus the channel network, resulting to a conjunction of four first-order debris flow channels deeply incised in the deposit down to the bedrock in several reaches. The upper catchment can be therefore subdivided in 5 subcatchments in terms of sediment recharge (Fig. 3). Bed entrenchment is now much constrained by check dams. However, lateral
erosion still occurs episodically by flooding and debris flow scouring.
   The style of sediment production and delivery is somehow different throughout the headwater, according to the local morphology and the lithologic and structural setting. The major geomorphic processes, identified preliminary in details from aerial photographs observations and field investigations, were characterized in a map (Fig. 4)
that describes the spatial distribution of geomorphic features and sediment transfer processes contributing to recharge the first-order channels with debris. The west and upper sides are dominated by rockfall. Large rock collapses delimited by persistent joints occur due to the progressive degradation of the slope underneath (Loye et al., 2011). Where the slope gradient allows scree and soil development, erosion scars can
be observed; sediment sources are remobilized from discrete shallow landslide. Depending on the location and size, rockfall reach the channels directly, or accumulate on slopes or in ravines, before being subsequently routed to high-order segments by a combination of gravitational and hydrological processes. Towards east, the erosion

## ESURFD

doi:10.5194/esurf-2015-48

**Headwater sediment
dynamics in debris
flow catchment**

A. Loye et al.

seems to be more progressive through the formation of gullies (Loye et al., 2012). The slopes near the ridge display mostly talus and scree deposits lightly covered with vegetation, whereas the hillside below exposes steepened rock slopes. Many active erosion scars can be observed. They contribute to accumulate debris into gullies and talus slope deposits that are subsequently entrained in channels downslope.

Historical records of debris flow since the 18th century show a frequency of 0.3 events per year that reached the apex of the fan (Brochot et al., 2000). The largest event deposited about $60\,000\,\mathrm{m}^3$. However, the torrent experiences smaller flux of debris ($< 1000\,\mathrm{m}^3$) usually not reported in archives. Such events can occur 2–3 times per year, when initiated by intense runoff (Veyrat-Charvillon, 2005). Volume of debris deposited in the sediment trap for the last 25 years represents $2200\,\mathrm{m}^3\,\mathrm{yr}^{-1}$, reaching a maximum of $7000\,\mathrm{m}^3\,\mathrm{yr}^{-1}$ in 2008 (RMT service).

# 3 Methods and data processing

## 3.1 Topographic monitoring using TLS

The terrain was surveyed with an ILRIS-3D laser scanner (Optech Inc.). This device provides a range up to 1.2 km for 80 % reflectivity surface and the instrumental precision is about 7 mm/100 m range for both distance and position (Optech Inc.). The entire coverage of the upper catchment with TLS point clouds required 50 scans considering a 20 % surface overlapping. They were collected over a 5 days survey from 9 individual viewpoints to ensure a 3-D rendering of the topography. A particular attention was carried in irregular regions and major break of slopes, such as rock couloirs and deep-cut gullies. Using multiple scanning locations allow to limit shadow zones and increase the point cloud density of the scanned area. A series of 4 surveys was performed on a seasonal basis during 2009 and one extra survey was performed in July 2010 to analyse the effect of the winter period (Table 1). The monitoring setup remained similar for all surveys. Post-processing of TLS raw data was done using Polyworks (InnovMetric). Er-

Discussion Paper | Discussion Paper | Discussion Paper | Discussion Paper |

**ESURFD**

doi:10.5194/esurf-2015-48

**Headwater sediment dynamics in debris flow catchment**

A. Loye et al.

roneous points and vegetation were filtered manually, ensuring a total control of the removed data to preserve a high density of points in topographic features with small radii curvature. Although this manner is time consuming, (semi-)automatic approaches to filter vegetation accurately still remain in a stage of development for dissected mountain morphology (Brodu and Lague, 2012). Each multiple scans of a survey were merged to one another using common tie points of permanent topographic features and set in 12 local subsets. Given the size of the monitored area, dividing the point cloud in smaller datasets allows to avoid propagation of inaccuracy through large co-registered scans series. ICP (iterative closest point) algorithms (Besl and McKay, 1992), that enable to minimize the distance between two sets of points, were used to determine the best alignment of a multiple scans subset in order to obtain the best co-registration within a time series. The same procedure was applied between the generated subset point cloud and a commercial airborne laser scanner derived point cloud (mean density: 6.9 pts m$^{-2}$) acquired in June 2009 to get the TLS data georeferenced in the Lambert projection coordinate system. The initial survey point cloud data was set as the surface model of reference. Each successive survey was best georeferenced on this reference using the ICP processing steps herein. The topographic change occurring between two successive surveys are too localized to influence the global co-registration within two survey data subsets consisting of millions of data points, hence the alignment accuracy. More details about multiple scans registration techniques and point cloud time series comparison can be found in Oppikofer (2009). The generated surface produced by a survey possesses a point spacing ranging from 2.5 to 18 cm according to the distance of acquisition. A maximum range of about 800 m was reached on the top peak of the catchment with a point cloud density of 25 pts m$^{-2}$. The surface coverage represents 84 % of the deforested area under investigation (Table 2).

## 3.2   Topographic changes identification and characterization

The active geomorphic features within two successive datasets were identified on a point to point approach using the short distance neighbouring point search algorithm

(Bitelli et al., 2004) that compute in 3-D the shortest difference vectors between points of two datasets. The vector sign indicates the net change direction of topography, i.e. surface of erosion or deposition. A set of points (cluster) was considered as active if at least 8 adjacent points of similar sign displayed an absolute difference above the limit of detection (LoD). Each active feature was outlined visually using the point cloud of difference (Fig. 5a). The point clusters of both survey datasets, which correspond to the topography of the active features, were extracted according to their spatial extend coordinates and each detected geomorphic feature was labelled:

1. Rock slope erosion characterises rockfall/-slide;

2. Hillslope erosion concerns the reworking of loose/compacted debris on slopes, respectively in gullies and channels;

3. Deposition delineates material aggradation initiated by both rock slope failure (new production) and remobilisation of debris.

Using the images captured by the TLS integrated camera, clusters of points not corresponding to geomorphic process activity, such as snow melt, were ignored.

### 3.3 Volume computation of each geomorphic feature

As the volume of active features cannot be directly computed from TLS datasets differencing, the active features of two successive point clouds must be interpolated into continuous surfaces (DEM). Gridded model (raster) is regarded as being most effective from irregularly distributed datasets containing in some parts only few or no point (El-Sheimy et al., 2005), as this can be the case for rockfall and erosion scar. The algorithm of interpolation has however minor impacts, as TLS data provide an extremely dense coverage of the detected objects (Anderson et al., 2005). So, they were interpolated using linear inverse distance weighting (Burrough and McDonnell, 1998) and generated in a regular grid separately. Grid spacing and direction of interpolation were

**ESURFD**

doi:10.5194/esurf-2015-48

**Headwater sediment dynamics in debris flow catchment**

A. Loye et al.

designed in a specific way for each feature: the coordinate system of reference was replaced by a local orthogonal system where the $x$-$y$ axes represent the average plane of topography nearby (Fig. 5b). This new reference frame was defined using eigen-value decomposition of the covariance matrix of the point cloud of reference (Shaw, 2003).

Interpolating the surface elevation in the direction of local topography allows generating a highly realistic DEM independently of slope steepness and thus, a close realistic representation of topography in case of overhanging features. The cell size was defined according to the point spacing distribution of both datasets. A series of tests revealed that setting the grid spacing at 68 % of the cumulative frequency distribution of point spacing provides a continuous surface reconstruction while keeping a high degree of detail from the point cloud. This ensures an accurate volume computation of geomorphic features. The volume was computed as the sum of the cell difference in elevation between the successive DEM. Absolute cell differences lying below a given threshold were not considered. The processing of volume computation using local deterministic method of interpolation adopting the adaptive gridding approach was developed in Matlab numerical computing environment.

## 3.4 Point cloud accuracy and limits of detection of the geomorphic features

A reliable identification of erosion and deposition features requires the setting of a minimum LoD, where the change of elevation between successive point clouds can be considered as real in opposition to noise associated with each dataset. Each TLS data point has theoretically a unique precision depending on the range and laser incidence angle (Buckley et al., 2008). In practice, the individual point precision of a scan can be assumed to model a surface with a global uniform uncertainty, considering the very high point density (Abellan et al., 2009). Given the homogeneity of surface error, and considering that the distance between sequential points at a position ($x$, $y$) should tend to zero, the accuracy of TLS data can be estimated by substituting the precision of each data point by a singular measurement of the error associated with the entire point distribution across the surface (Lane et al., 2003). Hence, the uncertainty related

## ESURFD

doi:10.5194/esurf-2015-48

**Headwater sediment dynamics in debris flow catchment**

A. Loye et al.

to both scans registration and point cloud georeferencing, the instrumental error included, was defined by the standard deviation of the distance ($\sigma_d$) between the points (Fig. 6). The LoD was therefore set at $2\sigma$ of the co-georeferencing and corresponds to the 95 % confidence limit (Table 3). Comparison with the approach considering the

5 error propagation for all uncertainties associated with each point cloud, and assuming a normal distribution of the error in distance (Taylor, 1997), shows that the uncertainties considered here are reliable.

In the case of volume computation, information on elevation uncertainty associated with each point cloud survey needs to be extended on a DEM cell by cell basis. For

any grid cell $(i, j)$ generated by the interpolation of adjacent points $p$ with independent elevation, the uncertainty of a cell elevation can be considered as the standard deviation ($\sigma_e$) of the data points elevation, where $\sigma_{\overline{e}_{i,j}} = \sigma_{e_p} / \sqrt{n}$ according to the equation of standard error of the mean, $n$ being the number of points to define the cell elevation. The elevation uncertainty for each cell in a DEM of difference is then expressed by:

$$\sigma_{\Delta\overline{e}_{i,j}} = \sqrt{\left(\sigma_{1\overline{e}_{i,j}}\right)^2 + \left(\sigma_{2\overline{e}_{i,j}}\right)^2}.$$

The volume uncertainty is then calculated by summing up the derived volume uncertainty of each cell of the feature as follow:

$$\Delta\overline{V}_{\text{feature}} = a \left[ \sqrt{\sum_{i=1}^{n}\sum_{j=1}^{n}\left(\sigma_{\Delta\overline{e}_{i,j}}\right)^2} \right], \text{ with } a = \text{cell area.}$$

The smallest detectable volume is about $10^{-3}\,\text{m}^3$ ($10\,\text{cm} \times 10\,\text{cm} \times 10\,\text{cm}$) (Table 3), but

can reach up to $0.006\,\text{m}^3$ ($25\,\text{cm} \times 25\,\text{cm} \times 10\,\text{cm}$) depending on the point spacing at maximum range. Topographic change detection and volume computation hang however not only on the quality of TLS data, such as point density and post-processing related inaccuracy. The complexity in surface geometry must be considered, like here,

Discussion Paper | Discussion Paper | Discussion Paper | Discussion Paper | Discussion Paper |

**ESURFD**

doi:10.5194/esurf-2015-48

**Headwater sediment dynamics in debris flow catchment**

A. Loye et al.

**ESURFD**

doi:10.5194/esurf-2015-48

**Headwater sediment dynamics in debris flow catchment**

A. Loye et al.

by integrating the range in position of all data points defining each grid cell value of a feature. Monitoring the hillslope activity is also limited by the ability of the process to create a distinct topographic change. Consequently, the deposition of individual small rockfall was not always detected, as detached rock mass fragments into pieces of sizes that are below the LoD. A similar issue was observed for erosion process on debris. Nevertheless, most of the material accumulation could be related to landslides or scouring. The sediment budgets were therefore kept in volumetric units, as they are commensurate for a consistent analysis. They were not converted to mass, although this would make more sense for comparing hillslope processes and rock slope yields. Such conversion requires an accurate density value of each surface process, whose approximations would bring unknown inaccuracies. Deposition related to rock failures may be slightly overvalued in the sediment balance, although parts of the volumetric amplification are compensated by a limited detection of small features.

## 3.5 Sediment budgets of the Manival torrent

The monitoring of the coarse sediment transfer was performed all along the main torrent channel to the sediment trap located downstream on the alluvial fan. The in-storage change was established after every noticeable flow event, using the morphological approach based on cross-section survey techniques (Ashore and Church, 1998), and the volume of sediment deposited in the sediment trap was measured by TLS survey differencing. Sequential volumes of recharge enable to study the influence of debris supply from the production zone through seasons. The characteristics and observational analysis of this event-based monitoring was documented in details in Theule et al. (2012, 2015) and is therefore only shortly reported here.

## 3.6 Estimation of debris production rate

A rate of debris production for the study period is obtained from the total volume of rock slope erosion. A more objective estimation can be deduced by characterising the

cumulative distribution of rockfall volumes with a power law as follows (Gardner, 1970):

$$N(v > V) = aV^{-b}.$$

$N$ is the rockfall frequency for a volume $v$ greater than $V$, $a$ and $b$ are constants. $a$ depends on the study size and on rock slope properties, whereas $b$ tends to be rather site independent (Dussauge-Peisser et al., 2002; Dewez et al., 2011). Considering that rock slope process activity causing rockfall does not fluctuate much over time, the inventory analysis can be used to infer the frequency of occurrence of larger events. This is done by integrating the rockfall frequency derivative $n(v) = \frac{dN}{dV}$ over the range of potential volumes. Estimation of the total volume $V_t$ per unit time that can be expected in average over a longer period of observation is therefore expressed by (modified from Hantz et al., 2002):

$$V_t = \int_{n(V_{min})}^{n(V_{max})} V \, dn = -ab \int_{V_{min}}^{V_{max}} V \times V^{-b-1} dV = -ab \int_{V_{min}}^{V_{max}} V^{-b} dV = \frac{-ab}{(1-b)} V^{1-b} \Big|_{V_{max}}^{V_{min}}$$

The goodness of fit of the power law was evaluated with the $\chi^2$ test (Taylor, 1997) and the standard deviation of value $a$ and $b$ were determined with the maximum likelihood estimate (Aki, 1965). The erosion rates are assessed by dividing $V_t$ with the surface prone to rockfall.

## 4   Results: hillslope process activity monitoring

### 4.1   1st monitoring period (April 2009–August 2009)

The topographic changes recorded from July to August 2009 showed any relevant geomorphic activity (only few small rockfall). The results were therefore merged with the preceding monitoring period.

ESURFD

doi:10.5194/esurf-2015-48

**Headwater sediment dynamics in debris flow catchment**

A. Loye et al.

**ESURFD**

doi:10.5194/esurf-2015-48

Rock slope activity is dominated by individual small rockfall disseminated throughout the upper catchment. Only few events exceed $1\,m^3$, such that contributions in terms of debris production are marginal in most parts of the catchment (Fig. 7). The significant geomorphic activity was located almost exclusively in the major gullies of Baure and Grosse Pierre ravines, and consists essentially of debris scouring of few $100\,m^3$ re-deposited further down. Material re-entrainment was also observed in several other smaller gullies, but their volumes are not relevant. The rock couloirs of the Genievre subcatchment and the scar of the old rock deposit showed barely any geomorphic activity. The channels displayed a net incision ($-636 \pm 43\,m^3$) in the upper reaches. Bedload aggradation remains very low ($+90 \pm 6\,m^3$). Below the upper confluence, the channel trunk exhibits a mixed pattern of zones of erosion ($-60 \pm 2\,m^3$) and deposition ($+80 \pm 4\,m^3$) induced by bedload transport. The flow path scours in-channel gravel-wedges and creates new depositions further.

## 4.2   2nd monitoring period (September 2009–November 2009)

Rock slope activity remains similar in spatial extend and volumes to the previous survey period, but rockfall frequency is higher (Fig. 8). Hillslope process activity was more widespread side east, but still particularly local on the western valley walls, while the rock couloirs showed any geomorphic activity. In the upper headwater, material reworking concentrated almost exclusively in steep tributary gullies. They displayed scouring of significant volume ($-357 \pm 12\,m^3$). Deposition features along the thalweg were quasi inexistent ($+18 \pm 1.3\,m^3$). In the south east, not only the Baure Ravine (net erosion: $-61 \pm 8\,m^3$), but the whole series of hillside gullies exhibited sign of activity, such as erosional segments alternate with deposition. On scree slopes, several minor rilling and their associated debris deposits were observed, some of them reached the channel trunk ($+42 \pm 2\,m^3$). Such small hillside debris flows were probably triggered by sediment entrainments in rills themselves, as no evidence of sliding at their head was observed. The channels show a net erosional character upstream ($-482 \pm 18\,m^3$), whereas continuous incisions were more pronounced in the Manival channel ($-443 \pm 16\,m^3$) as

**ESURFD**

doi:10.5194/esurf-2015-48

**Headwater sediment dynamics in debris flow catchment**

A. Loye et al.

in the Roche Ravine ($-40 \pm 3\,\mathrm{m}^3$ ). Deposition zones were almost completely absent ($15 \pm 1.3\,\mathrm{m}^3$). Towards the upper confluence, the lower segments of Manival channel exhibited continuous zones of aggradation ($97 \pm 6\,\mathrm{m}^3$) that were scoured on one side. This morphology is characteristic of close-process debris flow levees and run-up zones beside the incised channel bed. Below the upper confluence, channel bed cut ($-40 \pm 2\,\mathrm{m}^3$) and fill ($+16 \pm 1\,\mathrm{m}^3$) was sparse and concentrated at the junction with hillside gullies. Such pattern of bed reworking evidences the connectivity of the Baure gully series with the channel trunk.

### 4.3 3rd monitoring period (November 2009–July 2010)

This period showed an important increase of rock slope production, both in frequency and magnitude, resulting from the occurrence of large slope failures and enhanced rockfall activity locally, for instance in rock walls made of calcareous marl situated directly above the Manival ($2035 \pm 39\,\mathrm{m}^3$) and the Roche Ravine ($256 \pm 17\,\mathrm{m}^3$) channels (Fig. 9). Most of debris collapses supplied the channel directly; the rest was temporary deposited in breaks of slopes. The lower headwater part showed a great fluctuation as well (Genievre: $116\,\mathrm{m}^3$; Grosse Pierre: $145\,\mathrm{m}^3$). At the top of the Baure Ravine, $816 \pm 25\,\mathrm{m}^3$ of rock fragments contributed substantially to recharge the gully head. Below, debris infilling was continuously scoured. An $1170 \pm 18\,\mathrm{m}^3$ rockslide is responsible for the large channel infilling of the Manival subcatchment. Several other smaller rockfall contributed to the recharge of tributary gullies and scree hollows. In the Roche Ravine, debris deposits were sparse, because rockfall remained of low magnitude in average ($571$ events $< 1\,\mathrm{m}^3$), although frequency was high ($578$ events). The large debris infilling of the channel head was caused by two erosion scars in side gullies ($270 \pm 14$ and $65 \pm 4\,\mathrm{m}^3$). In the rock couloirs of the Genièvre subcatchment, a significant accumulation of material through landslides and rockfalls was observed (remnant volume: $204 \pm 13\,\mathrm{m}^3$), regarding that hillslope erosion represent $450\,\mathrm{m}^3$ ($\pm14$). In the Grosse Pierre Ravine, $343 \pm 17\,\mathrm{m}^3$ of debris were accumulated at the rock couloir

**ESURFD**

doi:10.5194/esurf-2015-48

**Headwater sediment dynamics in debris flow catchment**

A. Loye et al.

Discussion Paper | Discussion Paper | Discussion Paper | Discussion Paper | Discussion Paper |

outlet, recharging the scree slope above the channel head. In the Col du Baure, an important aggradation of the lower part of tributary gullies was observed (remnant volume: $+142 \pm 2\,\mathrm{m}^3$), resulting from material entrainment. Several debris slides were also detected on scree slopes, without any contact with the channel trunk.

5  The upper channel-reaches were clearly depositional, in consequence of large slope failures. The Manival channel showed a continuous zone of remnant accumulation of $948\,\mathrm{m}^3$ ($\pm 18$) in which a portion was carried along downstream as bedload. Towards the confluence, erosion dominated clearly ($-487 \pm 19\,\mathrm{m}^3$) against deposition ($+25 \pm 3\,\mathrm{m}^3$ ). In the Roche Ravine, a sustained erosion in the scar of the old rock deposit produced debris accumulation mostly on slope. But a landslide of $190 \pm 9\,\mathrm{m}^3$ reached the channel. Globally, aggradation was observed all along the channel head ($+148 \pm 18\,\mathrm{m}^3$) and scouring was sparse ($-65 \pm 4\,\mathrm{m}^3$). From the confluence downstream, the channel behaviour is dominantly erosional ($-97 \pm 4\,\mathrm{m}^3$) with almost any aggradation ($+3 \pm 0.3\,\mathrm{m}^3$).

## 4.4 Rock slope production inventory

Over the 16 months, 1866 rockfalls of volumes ranging from $10^{-4}$ to $10^3$ were recorded. This yields a total of $3575 \pm 30\,\mathrm{m}^3$ and an erosion rate of $3.1\,\mathrm{mm\,yr}^{-1}$ given the topographic surface area of rock faces. The inventory follows a power law (Fig. 10) with a 99 % confidence level for events larger than $3\,\mathrm{m}^3$ ($\chi^2$ value = 17.3). For events larger than $1\,\mathrm{m}^3$, the power law is accepted at the 95 % confidence level ($\chi^2$ value = 5.89). Both threshold volumes provide a $b$ value close to $0.81 \pm 0.06$. Considering volumes above $10\,\mathrm{m}^3$ only (25 events) gives a $b$ value of 0.76. Below $0.1\,\mathrm{m}^3$, the observed frequency deviated clearly from power law regime until the roll-over reaches a quasi-constant rate for the smallest volumes. According to the inventory, rockfall of more than $1\,\mathrm{m}^3$ are expected $153 \pm 11$ times per year in average. The largest event ($1170\,\mathrm{m}^3$) occurs every two years, and the one year return period rockfall is about $465\,\mathrm{m}^3$. Considering only class of volumes of the inventory, the rock slope production reaches a rate of $3678 \pm 210\,\mathrm{m}^3\,\mathrm{yr}^{-1}$ ($4 \pm 0.3\,\mathrm{mm\,yr}^{-1}$).

## 4.5 Torrent in-channel storage changes

Two debris flows of probably multiple surges and several remarkable bedload transport events were observed in the main torrent during the survey period (Theule et al., 2012). A debris flow occurred on the 25 August 2009, caused by a short duration rainstorm. The volume of sediment eroded in torrent ($5232 \pm 136\,\mathrm{m}^3$) coincides that redeposited in both the torrent itself and the sediment trap ($5072 \pm 125\,\mathrm{m}^3$), suggesting that the induced material entrainment involved what was stored in the torrent in great majority (Table 4). Sediment input from the headwater can be considered as marginal. Before that, any significant torrent activity was observed, despite series of rainfall of low to moderate intensity. In September 2009, a long period of moderate rainfall intensity caused material reworking induced by bedload transport all along the torrent. However, no sediment went supplying the sediment trap. A net gain of storage supplied by the headwater could be monitored. In October, a succession of low intensity rainfall induced some major sediment transport in the torrent that aggraded the sediment trap with at least $302 \pm 36\,\mathrm{m}^3$. The sediment budget indicates clearly a recharge of $229 \pm 31\,\mathrm{m}^3$, a transfer of debris that was stored mostly in the distal part of the torrent. Throughout the winter, a gradual incision was observed all along the torrent resulting from frequent period of low intensity rainfall conjugated to snowmelt. Due to maintenance (dredging), the sediment trap was disturbed and any reliable data was available. Any sign of competent activity was detected anyway. A new debris flow on 6 June deposited $3320 \pm 176\,\mathrm{m}^3$ in the sediment trap. This time, a certain supply from the headwater was observed ($\sim 270\,\mathrm{m}^3$). This event was followed by series of intense rainfall without much reworking in the distal part, suggesting that no competent transfer occurred anymore towards the torrent. The in-torrent storage changes and estimated recharge budgets are shown for each monitoring period in Fig. 11.

Discussion Paper | Discussion Paper | Discussion Paper | Discussion Paper | Discussion Paper

**ESURFD**

doi:10.5194/esurf-2015-48

**Headwater sediment dynamics in debris flow catchment**

A. Loye et al.

# 5 Synthese

The overall transfer dynamic, from debris source zone to the apex of the fan, is illustrated in Fig. 12. The volumes detected during the 16 months study period reveal a net export of $3378 \pm 361\,\mathrm{m}^3$ of sediment from the headwater to the main torrent (Table 5). The overall rock slope yield is $3575 \pm 30\,\mathrm{m}^3$, for a volume of erosion reaching $3129 \pm 150\,\mathrm{m}^3$ on the hillside and $1809 \pm 92\,\mathrm{m}^3$ in the channel complex. Volume of deposition, induced from both debris production and material reworking, yields a total volume of $5135 \pm 251\,\mathrm{m}^3$, whereas only $1382 \pm 56\,\mathrm{m}^3$ (27 %) concern the channel complex. In the main torrent, the sediment transfer was important ($\sim 20\,000\,\mathrm{m}^3$; net storage change $-4950 \pm 118\,\mathrm{m}^3$) and essentially related to the occurrence of two debris flow (Theule et al., 2012), depleting significantly the in-torrent sediment storage of the distal parts (entrainment zone). Material deposited in the sediment trap for the survey period yields $6075 \pm 45\,\mathrm{m}^3$. During the autumn, bedload transport of several hundreds of $\mathrm{m}^3$ acts as recharging the torrent all along.

In the spring-midsummer period, the hillside sediment budget yields a total rock slope production of $99 \pm 6\,\mathrm{m}^3$, for a volume of erosion of $-547 \pm 50\,\mathrm{m}^3$ and deposition of $+408 \pm 35$ (Table 5). This suggests that about $238 \pm 61\,\mathrm{m}^3$ of material supplied the channel complex, coming almost exclusively from material re-entrainment in gullies (Fig. 13). The sediment budget of the channels indicates a significant degradation in storage ($-487 \pm 44\,\mathrm{m}^3$), comprising large and continuous incisions ($-636 \pm 43\,\mathrm{m}^3$) in the upper reaches and material aggradation ($+149 \pm 11\,\mathrm{m}^3$) in the lower reaches resulting mostly from zones of transient re-deposition. This results a recharge of the torrent of $+726 \pm 103\,\mathrm{m}^3$ for this survey period.

During the late summer–autumn season, the total volume of hillside erosion is of $-640 \pm 27\,\mathrm{m}^3$, resulting particularly from a widespread scouring of tributary gullies located east and southeast of the headwater (Fig. 14). The total volume of rock slope production ($50 \pm 3\,\mathrm{m}^3$) and deposition ($+182 \pm 12\,\mathrm{m}^3$) remain low. Globally, the sediment budget indicates, that the hillslope contributed at recharging the channel reaches

Discussion Paper | Discussion Paper | Discussion Paper | Discussion Paper |

## ESURFD

doi:10.5194/esurf-2015-48

**Headwater sediment dynamics in debris flow catchment**

A. Loye et al.

ESURFD

doi:10.5194/esurf-2015-48

**Headwater sediment dynamics in debris flow catchment**

A. Loye et al.

with sediment for about $510 \pm 30\,\mathrm{m}^3$ (Table 5). The channels sediment budget yields $-522 \pm 20\,\mathrm{m}^3$ of erosion for $+127 \pm 13\,\mathrm{m}^3$ of deposition. This is characterized by bed-load reworking in both low-order and trunk channels, and a progressive transfer of $+904 \pm 51\,\mathrm{m}^3$ of material in the torrent.

During winter–spring 2010, a total volume of deposition of $+3163 \pm 147\,\mathrm{m}^3$ is recorded on the hillside, for a volume eroded of $-3129 \pm 150\,\mathrm{m}^3$. An important production of debris ($3424 \pm 89\,\mathrm{m}^3$) is observed (Table 5). The net sediment balance on the hillside yields to a supply of $+2203 \pm 187\,\mathrm{m}^3$ of sediment in channels, and the one for the channel complex indicates an increase of in-storage sediment of $+455 \pm 47\,\mathrm{m}^3$, according to a total volume of deposition of $1105 \pm 36\,\mathrm{m}^3$ and erosion of $651 \pm 29\,\mathrm{m}^3$ due to large portions of bed scouring in the downstream reaches. Sediment transfer to the torrent is $1749 \pm 199\,\mathrm{m}^3$ (Fig. 15).

# 6 Discussion

## 6.1 Debris supply through rock slope production

Debris production from rock walls degradation shows a strong seasonal pattern. The great majority of recorded rock instabilities in both magnitude (95 %) and frequency (75 %) occurred during the cold period. Previous studies of the calcareous cliffs near Grenoble, which represent a similar morphotectonic context, revealed that freeze–thaw cycles are the main triggering factor of rockfall (Frayssines et al., 2006). Ice jacking can cause microcracks propagation leading to failure (Matsuoka and Sakai, 1999). Along the eastern ridge, the bedrock surface is often highly fractured, suggesting frost shattering. The spatial pattern of rockfall strongly suggests also a lithologic influence that can be explained by differential erosion between the successive limestone and marl beds. In the rock walls series side west, the monoclinal configuration of the bedding, combined with a strong difference of competency between stratigraphic sequences, give rise to overhanging formation highly susceptible to failure. On the east side, the

Discussion Paper | Discussion Paper | Discussion Paper | Discussion Paper |

bedding is mostly cataclinal and approaches dip-slope, depending on the slope. Rock failures initiated by planar sliding on bedding planes were observed.

The observed debris production follows a power law distribution in a range covering at least 3 orders of magnitude [$10^0$–$10^3$]. The exponent $b$ is slightly higher than the average value reported for the Grenoble cliffs [0.4–0.7] (Hantz et al., 2011), but is in agreement with short inventories covering a lower range of volume [$10^{-2}$–$10^2$] (Hungr et al., 1999; Dussauge et al., 2003). Inventories dominated by small volumes tend to increase the $b$ value, compared to the ones covering rather large volumes (Stark and Hovius, 2001). Above $100\,\mathrm{m}^3$, the deviation from the power law may be attributed to the short period of sampling for events of such large magnitude. The roll-over encountered towards small volumes results most likely in the under-detection of the number of events. This sampling bias being far above the minimum volume of detection ($0.006\,\mathrm{m}^3$), another behaviour characterizing the failure of small volumes cannot be excluded. This may presuppose a physical erosion process that differs from the one influencing larger instabilities, which are controlled primarily by local predisposing factors, such as the geometrical and geomechanical properties of the rock mass (Selby, 1993; Sauchyn et al., 1998), and such as the local conditions of tectonic weakening (Cruden, 2003; Coe and Harp, 2007). Low magnitude rockfall represent anyway little contribution in terms of debris supply, even though they vary locally from 1 or 2 orders of magnitude in volume over the time, as observed here. The amount of sediment available is only significantly influenced by instabilities of high magnitude (Fig. 16).

Previous sediment budgets derived from topographic measurement using stereophoto-grammetry estimated the highest erosion rates over an average of 40 years to range from 10.8 to $17.8\,\mathrm{mm\,yr^{-1}}$ in the headwater (Veyrat-Charvillon and Memier, 2006). Although the large uncertainty of the approach, and the fact that they measured the hillslope and thalweg geomorphic activity, these values are compatible with the erosion rate revealed here from the short period rockfall inventory, by supposing the possible occurrence of rockslides of magnitude [$10^6$–$10^7$]. Considering that the power law is valid for larger slope failures, a $7500\,\mathrm{m}^3$ event can be expected every 10,

# ESURFD

doi:10.5194/esurf-2015-48

**Headwater sediment dynamics in debris flow catchment**

A. Loye et al.

**ESURFD**

doi:10.5194/esurf-2015-48

**Headwater sediment dynamics in debris flow catchment**

A. Loye et al.

respectively 100 years for 120 000 m$^3$. The average debris production ranges between 5587 ± 241 to 12 903 ± 305 m$^3$ yr$^{-1}$, according to a maximum potential erosion of 10$^5$, respectively 10$^7$ m$^3$, over several centuries (Table 6). Any historical Manival rockslide exists to support this estimation. The large old rock deposit ($\sim$ 6.1 Mm$^3$) of the upper catchment is the largest detected event, but it may have formed from several rock collapses. Rockfall inventory of the Grenoble cliffs reports volume smaller than 10$^5$ m$^3$ for the last century, respectively 10$^7$ m$^3$ since the 17th century (Hantz et al., 2003). Such a magnitude is also likely at Manival. A mean rate of rock slope erosion of about 10 mm yr$^{-1}$. (10 000 m$^3$ yr$^{-1}$) can be therefore expected in the upper catchment over the century.

Upstream from the Manival channel, scouring of debris slopes and scree hollows induced from rock slope production contributed for instance for about 40 % of the net erosion recorded during the autumn period, respectively 25 % in the Baure Ravine over the entire study period. The dominant mode of debris supply in the Manival headwater is therefore highly episodic, implying a great spatial heterogeneity in recharge rate.

## 6.2   Debris supply through hillslope activity

Rock slope activity was very little from spring to autumn, such as hillslope geomorphic activity dominated the process of sediment recharge. Until the end of August, ephemeral debris-filled tributary streams and hillside gullies remain quasi inactive in terms of sediment delivery. The geomorphic activity of both hillside and low-order channels recorded the lowest transfer of material on average. The autumn period is characterized by a general increase in intensity of geomorphic activity. The negative sediment balances in all sediment cascade components suggests a very high degree of connectivity between hillside and low-order channels. Continuous scouring and the quasi inexistence of deposition features from hillside gullies indicate that mobilized material is integrally entrained downstream by runoff. In the channel reaches, clear incisions and micro debris flow indicate a competent transfer of material. Globally, the

**ESURFD**

doi:10.5194/esurf-2015-48

**Headwater sediment dynamics in debris flow catchment**

A. Loye et al.

hillside contribution represents on average a volume 5 times larger than what was observed in spring and summer. Channel bed reworking was of much larger magnitude as well. During winter-spring 2010, the huge increase of debris production can be attributed to the winter, according to observations carried out in the preceding spring. Consequently, the total volume of deposition recorded on the hillside increased readily (on average $382 \pm 18 \, \text{m}^3 \, \text{month}^{-1}$), which overcomes significantly the rate of deposition recorded so far. Hillslope and gully erosion remain in average comparable to the volumetric transfer of sediment observed in the preceding autumn, implying however a clear connectivity. Hillside fan deposits observed along low-order channel banks reflect thus an effective hillslope-channel coupling, whereas local sediment budgets are dominated by a recharge of material.

### 6.3  Sediment recharge of the torrent

The sediment input, back-calculated from the in-torrent storage changes, coincides with the net sediment output recorded from the headwater for the first two survey periods. In the torrent, the morphological monitoring that started in July revealed quasi no recharge ($< 70 \, \text{m}^3$) and is coherent with observations made in summer in the upper catchment. The headwater sediment output must have accumulated before, probably mobilized as bedload by common runoff events in spring. In autumn, both budgets coincide ($1018 \pm 84 \, \text{m}^3$ against $904 \pm 51 \, \text{m}^3$), considering that few segments between both entities are missing, and that both budgets were in volumetric units, although different process density. The morphological budget indicates that the torrent experienced a net recharge in the distal part, suggesting that the upper reaches acted as an effective buffer to the transfer of sediment from the headwater. This enhanced debris supply in torrent results from the efficient transport of material from sediment production zone (headwaters) and all along the sediment network components, and emphasizes the high linkage from the source zones to the torrent. In the 3rd survey period, the headwater sediment balance indicates a net export of debris ($1749 \pm 199 \, \text{m}^3$), whereas the morphological monitoring detected any significant volume of debris entering the main

**ESURFD**

doi:10.5194/esurf-2015-48

torrent. Even the recharge (sediment input in Fig. 11) measured during the June debris flow events ($< 600\,\mathrm{m}^3$) remains far below the transfer of sediment recorded upstream in the headwater. This discrepancy may result from material deposition right in the non-monitored segments at the headwater outlet. But field study did not confirm any

of such trace. The analysis of past series of sediment budgets performed in the upper Manival catchment (Veyrat-Charvillon, 2005) reveals, that the spring-early summer time exhibits currently a period of recharge following a phase of discharge within a short time lapse depending on the hydrometeorological and snow melt conditions. The most reasonable explanation results therefore in the long time interval between measure-

ments, such as the successive reworking of bedload transport must have inhibited the cut and fill pattern, and masked the short term behaviour of sediment transfer operated in the torrent. This is a well-known issue when working with channelized hillslope processes (Fuller et al., 2010). Although this monitoring aspect concerns the topographic changes recorded by TLS in the headwater as well, geomorphic activity, such as micro

debris flows and continuous channel bed degradation, strongly suggest a net sediment delivery toward the torrent that must have occurred before the debris flow event.

## 6.4 Possible causes of seasonal fluctuations in debris supply

Headwaters are currently underlined as being more sensitive to intense rainfall in terms of geomorphic activity (Veyrat-Charvillon, 2005; Brayshaw and Hassan, 2009)

and, therefore, should be more sediment delivery responsive than higher-order channels. However, the Manival headwater experienced low geomorphic activity through the summer, and consequently low recharge of the torrent, although the occurrence of high intensity rainstorms competent enough to trigger debris flow of significant magnitude in torrent. Considerations of the temporal pattern of sediment transfer and the analysis of

erosion features, like alternating areas of scouring and infilling in gullies, suggest that runoff still exerts an important role on the headwater sediment dynamics. A clear relation between sediment transfer magnitude and precipitation remains complex however. The enhanced hillside geomorphic activity observed in several headwater subsystems,

**ESURFD**

doi:10.5194/esurf-2015-48

**Headwater sediment dynamics in debris flow catchment**

A. Loye et al.

for instance during the autumn period, induced simultaneously a highly heterogeneous response in their channel reaches; a significant increase of bed incision and debris flow similar reworking was observed in the upper reaches of the Manival subcatchment, implying an important sediment transfer. In contrast, the activity of other channel reaches,

for instance Roche Ravine, was reduced by half.

Considering that meteorological conditions were similar, this opposite behaviour can only be explained by a certain starvation of debris availability. During the entire preceding study period, any particular event was observed in the Roche Ravine, which remained geomorphically much inactive. Only little sediment recharge was observed.

In the consequence, the channel segments underwent sediment exhaustion at some point, implying a reduced sediment yield. Exhaustion comes not only within a supply-limited regime of the contributing area, but also from the fact that check dams, like bedrock dominated reaches, inhibit channel bed incision. Hence, the sediment storage has to be refilled either from the contributing hillside or from upstream mass move-

ment. A similar observation can be drawn from the Grosse Pierre Ravine sediment budget, whose gully downslope remained completely disconnected from the head over the whole study period at least. Although this ravine is very steep and incises the large old rock deposits, no geomorphic work was observed, resulting most likely from the absence of debris supply from upstream. Hillside sediment delivery seems therefore to

be clearly a limiting factor to sediment yield from low to high-order channels, and thus to the recharge rate of the debris flow torrent downstream. The Genièvre subsystem, which showed to be mostly inactive in 2009, displayed a large sediment transfer in spring 2010 until exhaustion. As the occurrence of bedload transport and micro debris flow is controlled predominantly by the availability of mobilizable sediment, numerous

suitable meteorological conditions do not conduct systematically to significant transfer of sediment from the hillside to low-order channels.

## 6.5 Role of sediment source stability

The sediment delivery from the hillside was very low in summer, but increased significantly during autumn, meaning that debris storage was not depleted. This behaviour is somehow equivocal, considering the fact that the transport capacity of ephemeral

stream runoff and sheetwash related to high intensity rainstorms are larger than the one generated by low intensity long duration rainfall; above all, when gully material (like in Manival) can be characterized as coarse and poorly sorted rockfall fragment derived debris. The role of long lasting rainfall in sediment transfer can be related to the stability of the coarse surface layer armouring the gullies and scree slopes. Excess pore-fluid

pressure in debris deposits can persist for days to weeks after sediment emplacement time (Major and Iverson, 1999; Major, 2000). One possible explanation is, that the 25 August rainstorm dramatically altered the debris sources in a way that the autumn rainfalls, although of lower intensity but longer flood time, were able to transfer sediment downslope, for instance by saturating the debris deposits on the long term. This sug-

gests, though, that periods of successive wetter meteorological conditions can enhance sediment transfer due to more unstable sediment sources. The spatial pattern of geomorphic work showed, that hillslope process activity was observed principally in gullies and scree slopes situated directly below active rock walls. As expected, fresh rock deposits are more easily destabilized than old compacted colluvium. This is supported

by the rapid sediment exhaustion observed in rock couloirs for instance. The recurrent supply of fresh sediment seems especially prone to be transported downstream. Lenzi et al. (2003), for instance, interpreted the annual fluctuation in sediment yield as the effect of sediment source destabilization or reactivation following a high-magnitude flow event, which facilitates material entrainment by subsequent runoff. Johnson and War-

burton (2006) refer to the influence of sediment source characteristic in the control of hillslope sediment discharge. These observations yield the impression, that sediment source stability affects the amount of debris supply from the hillside, despite flow ca-

Discussion Paper | Discussion Paper | Discussion Paper | Discussion Paper |

# ESURFD

doi:10.5194/esurf-2015-48

**Headwater sediment dynamics in debris flow catchment**

A. Loye et al.

**ESURFD**

doi:10.5194/esurf-2015-48

**Headwater sediment dynamics in debris flow catchment**

A. Loye et al.

pacity and sediment availability, and that fresh debris deposits are usually less stable geotechnically.

## 7  Conclusion

This investigation of a yearly cycle of sediment dynamic, by evaluating sediment
sources and transfer processes, could underline that both debris flows displayed an identical behaviour in pre-event debris supply: phases of recharge preceded in time their occurrence, which could be of several months or even years. The rate of sediment delivery, recharging both hillside and low-order channels directly, was essentially caused by high magnitude slope failure of moderate frequency and occurred mostly
during winter time. Consequently, material re-entrainment concentrates locally in specific tributary gullies. As potential instabilities are widespread in the head walls (Loye et al., 2011), a substantial recharge is likely each winter. But the location and magnitude of sediment supply would be probably much different in a different year.

    During this study period, the seasonal cycle of sediment discharge from the head-
water supplying the torrent with debris consisted of two phases of recharge:

1. in early spring, linked to runoff conditions and probably wet snow avalanches.

2. in autumn during long period of rainfalls and wet meteorological conditions.

In the Manival, the response in terms of debris supply seems therefore to be directly subordinated to the onset of a period of runoff derived from rainfall or snowmelt, debris
availability being usually not a limiting factor. Conversely, the effect of intense summer rainstorms is limited by the fact that headwater subsystems follow a sediment supply–exhaustion–supply cycle, as a corollary to the supply-limited model of debris flow initiation (Jakob et al., 2005; Glade, 2005). In the case of sediment depletion, as low-order channels are supply-limited, even a very competent rainstorm derived
runoff will not deliver sediment to the torrent. The time rate of exhaustion seems to be

**ESURFD**

doi:10.5194/esurf-2015-48

**Headwater sediment dynamics in debris flow catchment**

A. Loye et al.

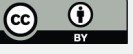

highly local, depending not only on debris availability, but probably on geomorphological aspect (Mao et al., 2009). The sediment volume temporary accumulated in local topographic hollows depends on the degree of convergence and on the slope gradient for instance (Reneau et al., 1990; Stock and Dietrich, 2006). And the state conditions

of sediment source stability in gullies and scree hollows in particular can influence sediment retention in the headwater. The destabilizing effect can be related to hydrometeorological conditions since the last rainstorm, rather than to flow capacity directly. Globally, the torrent effectiveness seems to be controlled early in the season by sediment production. and later in the season by the ability of hydrological effects to weaken

the remnant debris sources. Low-order reaches contribute significantly to the sediment delivery mechanism of the catchment headwater, by controlling storage and routing of debris in the drainage system. Hence, the recharge threshold required for a new debris flow to occur depends primarily on the short-term debris supply, partly derived from the rate of rock slope sediment production and partly derived from the availabil-

ity of mobilizable debris on the hillside. The rate of sediment recharge in torrent is however greatly inconstant, since production and entrainment are both highly stochastic processes. This regime of headwater sediment delivery may have been identified in other close mountainous environments, but very little literature exists (Alvarez and Garcia Ruiz, 2000; Veyrat-Charvillon, 2005; Berger et al., 2011), that have explored in

sufficient detail the time scale of sediment discharge on a seasonal basis.

Debris flow magnitudes have been mostly determined so far based on volume estimates derived from past events, reducing the susceptibility analysis to the known history. A monitoring of the in-storage changes in torrent linked to the debris supply can readily improve knowledge on recharge threshold leading to debris flow activity, and

25 therefore their prediction. $10\,000\,\mathrm{m}^3\,\mathrm{yr}^{-1}$ of debris supplying the headwater channels can be expected in Manival over a century, according to the rock slope production observed in this study. Although the multiplicity of sediment sources and mode of transfer operating at different spatial and temporal scales, the pattern of processes governing the sediment dynamic can be considered precisely on a seasonal basis using TLS

techniques. The maximum discharge that can initiate an extreme event that would mobilize the entire torrent storage could be specified. Without direct measurement of the rate of sediment flux and of the coupling between hillslope and channel processes, this cannot be rigorously determined. The timing of sediment budget monitoring in channels is however a crucial aspect for their interpretations.

*Acknowledgements.* The authors would like to thank their colleagues at IGAR and IRSTEA Grenoble (ex. CEMAGREF), in particular A. Pedrazzini and M.-H. Derron for their valuable comments during the preparation of this publication. This study was funded entirely by the University of Lausanne, except for the event-based cross-section surveys that was funded by the Pôle Grenoblois d'étude et de recherche pour la prévention des risques naturels. The ONF-RTM38 is acknowledged for making the access to the upper Manival Catchment easier.

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

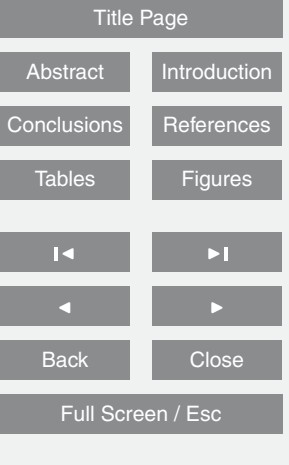

## ESURFD

doi:10.5194/esurf-2015-48

**Headwater sediment dynamics in debris flow catchment**

A. Loye et al.



Brayshaw, D. and Hassan, M. A.: Debris flow initiation and sediment recharge in gullies, Geomorphology, 109, 122–131, 2009.

Bremer, M. and Sass, O.: Combining airborne and terrestrial laser scanning for quantifying erosion and deposition by a debris flow event, Geomorphology, 138, 49–60, 2011.

Berti, M., Genevois, R., LaHusen, R., Simoni, A., and Tecca, P. R.: Debris flow monitoring in the acquabona watershed on the Dolomites (Italian Alps), Phys. Chem. Earth Pt. B, 25, 707–715, 2000.

Bitelli, G., Dubbini, M., and Zanutta, A.: Terrestrial laser scanning and digital photogrammetry techniques to monitor landslide bodies, in: Proceedings of the XXth ISPRS Congress Geo-Imagery Bridging Continents, XXXV, part B5, Istanbul, Turkey, 12–23 July 2004, 246–251, 2004.

Bovis, M. J. and Jakob, M.: The role of debris supply conditions in predicting debris flow activity, Earth Surf. Proc. Land., 24, 1039–1054, 1999.

Brochot, S., Coeur, D., Lang, M., and Naulet, R.: Historique – Isère et torrents affluents. Utilisation de l'information historique pour une meilleur définition du risque d'inondation (rapport), Cemagref/Achtys, Lyon, Grenoble, 248 pp., 2000.

Brodu, N. and Lague, D.: 3D terrestrial lidar data classification of complex natural scenes using a multi-scale dimensionality criterion: applications in geomorphology, ISPRS J. Photogramm., 68, 121–134, 2012.

Buckley, S., Howell, J., Enge, H., and Kurz, T.: Terrestrial laser scanning in geology: data acquisition processing and accuracy considerations, J. Geol. Soc. London, 165, 625–638, 2008.

Burrough, P. and McDonnell, R.: Principals of Geographic Information Systems, Oxford University Press, 333 pp., Oxford, 1998.

Cannon, S. H., Gartner, J. E., Parret, C., and Parise, M.: Wildfire-related debris flow generation through episodic progressive sediment-bulking process, western USA, in: Debris-Flow Hazard Mitigation: Mechanics, Prediction, and Assessment, edited by: Rickenmann, D. and Chen, L., Millpress, Rotterdam, 71–82, 2003.

Chandler, J. H. and Brunsden, D.: Steady state behaviour of the Black Ven mudslide: the application of archival analytical photogrammetry to studies of landform change, Earth Surf. Proc. Land., 20, 255–275, 1995.

Charollais, J., Dondey, D., Ginet, C., Lombard, A., Muller, J. P., Rosset, J., and Ruchat, C.: Carte géol. France (1/50.000°), Feuille Domène (33–34), B. R. G. M., Orléans, 1986.

**ESURFD**

doi:10.5194/esurf-2015-48

**Headwater sediment dynamics in debris flow catchment**

A. Loye et al.

Coe, J. A. and Harp, E. L.: Influence of tectonic folding on rockfall susceptibility, American Fork Canyon, Utah, USA, Nat. Hazards Earth Syst. Sci., 7, 1–14, doi:10.5194/nhess-7-1-2007, 2007.

Coe, J. A., Whitney, J. W., and Harrington, C. D.: Photogrammetric analysis of Quaternary hillslope erosion at Yucca Mountain, Nevada, Geol. Soc. Am. Astr. Progr., 25, 22, 1993.

Cruden, D. M.: The shapes of cold, high mountains in sedimentary rocks, Geomorphology, 55, 249–261, 2003.

Decaulne, A. and Saemundsson, P.: Spatial and temporal diversity for debris-flow meterotological control in subarctic oceanic periglacial environments in Iceland, Earth Surf. Proc. Land., 32, 1971–1983, 2007.

Dewez, T. and Rohmer, J.: Probabilistic rockfall hazard: empirical computation based on ground-based lidar observations in Mesnil-Val, Normandy, Journée Aléa Gravitaires, 7–8 September 2011, Strasbourg, France, 104–115, 2011.

Dietrich, W. F. and Dunne, T.: Sediment budget for a small catchment in mountainous terrain, Z. Geomorphol. Supp., 29, 191–206, 1978.

Dussauge-Peisser, C., Helmstetter, A., Grasso, J.-R., Hantz, D., Desvarreux, P., Jeannin, M., and Giraud, A.: Probabilistic approach to rock fall hazard assessment: potential of historical data analysis, Nat. Hazards Earth Syst. Sci., 2, 15–26, doi:10.5194/nhess-2-15-2002, 2002.

Dussauge, C., Grasso, J.-R., and Helmstetter, A.: Statistical Analysis of Rock Fall Volume Distributions: implications for Rock Fall Dynamics, J. Geophys. Res.-Sol. Ea., 108, 2–11, 2003.

El-Sheimy, N., Valeo, C., and Habib, A.: Digital Terrain Modeling: Acquisition, Manipulation, and Applications, Artech House, Boston, MA, 257 pp., 2005.

Frayssines, M. and Hantz, D.: Failure mechanisms and triggering factors in calcareous cliffs of the Subalpine Ranges (French Alps), Eng. Geol., 86, 256–270, 2006.

Fuller, I. C. and Marden, M.: Rapid channel response to variability in sediment supply: cutting and filling of the Tarndale Fan, Waipaoa catchment, New Zealand, Mar. Geol., 270, 45–54, 2010.

Gardner, J. S.: Rockfall: a geomorphic process in high mountain terrain, The Albertan Geographer, 6, 15–20, 1970.

Gidon, M.: Géologie de la Chartreuse – Sentiers de la Chartreuse: Circuit de la Dent de Crolles, Association "A la découverte du Patrimoine de Chartreuse", publ. 1d, 1. éd., p. 20, 9 fig., available at: www.Geol-alp.com (last access: 18 April 2014), 1991.

Discussion Paper | Discussion Paper | Discussion Paper | Discussion Paper |

**ESURFD**

doi:10.5194/esurf-2015-48

**Headwater sediment dynamics in debris flow catchment**

A. Loye et al.

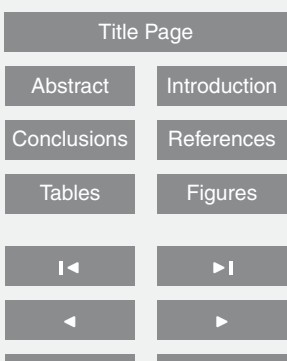

Glade, T.: Linking debris-flow hazard assessments with geomorphology, Geomorphology, 66, 189–213, 2005.

Gruffaz, F.: Torrent du Manival (Isère), Etude de basin et de la plage de dépôts torrentiels (rapport), RTM/ONF, Grenoble, 66 pp., 1997.

Guzzetti, F., Peruccacci, S., Rossi, M., and Stark, C. P.: The rainfall intensity–duration control of shallow landslides and debris flows: an update, Landslides, 5, 3–17, 2008.

Hantz, D., Dussauge-Peisser, C., Jeannin, M., and Vengeon, J.-M.: Rock fall hazard: from expert opinion to quantitative evaluation, Symposium "Geomorphology: from expert opinion to modeling (2002)", April 2002, Strasbourg, France, 115–122, 2002.

Hantz, D., Vengeon, J. M., and Dussauge-Peisser, C.: An historical, geomechanical and probabilistic approach to rock-fall hazard assessment, Nat. Hazards Earth Syst. Sci., 3, 693–701, doi:10.5194/nhess-3-693-2003, 2003.

Hantz, D.: Quantitative assessment of diffuse rock fall hazard along a cliff foot, Nat. Hazards Earth Syst. Sci., 11, 1303–1309, doi:10.5194/nhess-11-1303-2011, 2011.

Hooke, J.: Coarse sediment connectivity in river channel systems: a conceptual framework and methodology, Geomorphology, 56, 79–94, 2003.

Hungr, O.: Characterizing debris flows for design of hazard mitigation, in: Proceedings of the 5th International Conference on Debris Flow Hazards Mitigation/Mechanics, Prediction, and Assessment, Padua, Italy, 14–17 June 2011, edited by: Genevois, R., Hamilton, D. L., and
Prestininzi, A., Italian Journal of Engineering Geology and Environment, Book of abstracts, Keynote lecture, p. 5., 2011.

Hungr, O., Evans, S. G., and Hazzard, J.: Magnitude and frequency of rockfalls along the main transportation corridors of southwestern British Columbia, Can. Geotech. J., 36, 224–238, 1999.

Hungr, O., McDougall, S., and Bovis, M.: Entrainment of material by debris flows, chapt. 7, in: Debris Flow Hazards and Related Phenomena, edited by: Jakob, M. and Hungr, O., Springer Verlag, Heidelberg, Germany, in association with Praxis Publishing Ltd, 2005.

Jaboyedoff, M., Oppikofer, T., Abellan, A., Derron, M. H., Loye, A., Metzger, R., and Pedrazzini, A.: Use of lidar in landslide investigations: a review, Nat. Hazards, 61, 5–28, 2012.

Jakob, M., Bovis, M., and Oden, M.: The significance of channel recharge rates for estimating debris-flow magnitude and frequency, Earth Surf. Proc. Land., 30, 755–766, 2005.

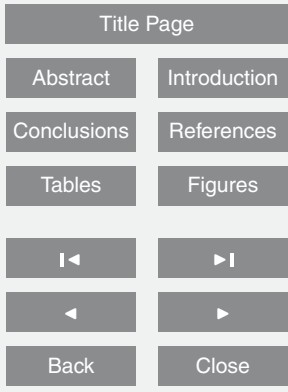

**ESURFD**

doi:10.5194/esurf-2015-48

**Headwater sediment dynamics in debris flow catchment**

A. Loye et al.

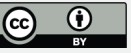

Johnson, R. M. and Warburton, J.: Variability in sediment supply, transfer and deposition in an upland torrent system: Iron Crag, northern England, Earth Surf. Proc. Land., 31, 844–861, 2006.

Johnson, R. M., Warburton, J., Mills, A. J., and Winter, C.: Evaluating the significance of event and post-event sediment dynamics in a first order tributary using multiple sediment budgets, Geogr. Ann. A, 92, 189–209, 2010.

Lane, S. N., Westaway, R. M., and Murray Hicks, D.: Estimation of erosion and deposition volumes in a large, gravel-bed, braided river using sysnotpic remoe sensing, Earth Surf. Proc. Land., 28, 249–271, 2003.

Lenzi, M. A., Mao, L., and Comiti, F.: Interannual variation of suspended sediment load and sediment yield in an alpine catchment, Hydrolog. Sci. J., 48, 899–915, doi:10.1623/hysj.48.6.899.51425, 2003.

Loye, A., Jaboyedoff, M., Pedrazzini, A., Theule, J., Liébault, F., and Metzger, R.: Morphostructural analysis of an alpine debris flows catchment: implication for debris supply, in: Proceedings of the 5th International Conference on Debris Flow Hazards Mitigation/Mechanics, Prediction, and Assessment, edited by: Genevois, R., Hamilton, D. L., and Prestininzi, A., Padua, Italy, 7–11 June 2011, Italian Journal of Engineering Geology and Environment, 115–126, doi:10.4408/IJEGE.2011-03.B-014, 2011.

Loye, A., Jaboyedoff, M., Pedrazzini, A., Theule, J., Liébault, F., and Metzger, R.: Influence of bedrock structures on the spatial pattern of erosional landforms in small alpine catchments, Earth Surf. Proc. Land., 37, 1407–1423, 2012.

Major, J. J.: Gravity-driven consolidation of slurries – implications for debris-flow deposition and deposit characteristics, J. Sediment. Res., 70, 64–83, 2000.

Major, J. J. and Iverson, R. M.: Debris-flow deposition – effects of pore-fluid pressure and friction concentrated at flow margins, Geol. Soc. Am. Bull., 111, 1424–1434, 1999.

Mao, L., Cavalli, M., Comiti, F., Marchi, L., Lenzi, M. A., and Aratto, M.: Sediment transfer processes in two Alpine catchments of contrasting morphological settings, J. Hydrol., 364, 88–98, 2009.

Oppikofer, T.: Detection, Analysis and monitoring of slope movements by high-resolution digital elevation models, PhD thesis, Inst. of Geomatics and Risk Analysis, University of Lausanne, Lausanne, 2009.

Matsuoka, N. and Sakai, H.: Rockfall activity from an alpine cliff during thawing period, Geomorphology, 28, 309–328, 1999.

# ESURFD

doi:10.5194/esurf-2015-48

## Headwater sediment dynamics in debris flow catchment

A. Loye et al.

Peiry, J. L.: Les torrents de L'Arve: dynamique des sédiments et impact de l'aménagement des bassins versants sur l'activité torrentielle, Rev. Geogr. Alp., 78, 25–58, 1990.

Perroy, R. L., Bookhagen, B., Asner, G. P., and Chadwick, O. A.: Comparison of gully erosion estimates using airborne and ground-based LiDAR on Santa Cruz Island, California, Geomorphology, 118, 288–300, 2010.

Péteuil, C., Maraval, C., Bertrand, C., and Monier, G.: Torrent du Manival: Schéma d'aménagement et de gestion du basin versant contre les crues, techn. report (unpubl.), Office National des Forêts, Service de Restauration des terrains en Montagnes de l'Isère, Grenoble, France, 107 pp., 2008.

Reneau, S. L., Dietrich, W. E., Donohue, D. J., Jull, A. J. T., and Rubin, M.: Late Quaternary history of colluvial deposition and erosion in hollows, central California Coast Ranges, Geol. Soc. Am. Bull., 102, 969–982, 1990.

Rickenmann, D.: Empirical relationship for debris flows, Nat. Hazards, 19, 47–77, 1999.

Roering, J. J., Stimely, L. L., Mackey, G. H., and Schmidt, D. A.: Using DInSAR, airborne LIDAR and archival air photos to quantify landsliding and sediment transport, Geophys. Res. Lett., 36, L19402, doi:10.1029/2009GL040374, 2009.

Sauchyn, D. J., Cruden, D. M., and Hu, X. Q.: Structural control of the morphometry of open rock basins, Kananaski region, Canadian Rocky Mountains, Geomorphology, 22, 313–324, 1998.

Schlunegger, F., Badoux, A., McArdell, B. W., Gwerder, C., Schnydrig, D., Rieke-Zapp, D., and Molnar, P.: Limits of sediment transfer in an alpine debris-flow catchment, Illgraben, Switzerland, Quaternary Sci. Rev., 28, 1097–1105, 2009.

Selby, M. J.: Hillslope Material and Processes, Oxford Univ. Press, Oxford, 451 pp., 1993.

Shaw, P. J. A.: Multivariate Statistics for the Environmental Sciences, Hodder-Arnold, London, ISBN: 0-3408-0763-6, 2003.

Smith, L. C., Alsdorf, D. E., Magilligan, F. J., Gomez, B., Mertes, L. A. K., Smith, N. D., and Garvin, J. B.: Estimation of erosion, deposition and net volumetric change caused by the 1996 Skeidararsandur Jökulhlaup, Iceland, from SAR interferometry, Water Resour. Res., 36, 1583–1594, 2000.

Stark, C. P. and Hovius, N.: The characterization of landslide size distributions, Geophys. Res. Lett., 28, 1091–1094, 2001.

Stock, J. D. and Dietrich, W. E.: Erosion of steepland valleys by debris flows, Geol. Soc. Am. Bull., 118, 1125–1148, 2006.

**ESURFD**

doi:10.5194/esurf-2015-48

**Headwater sediment dynamics in debris flow catchment**

A. Loye et al.

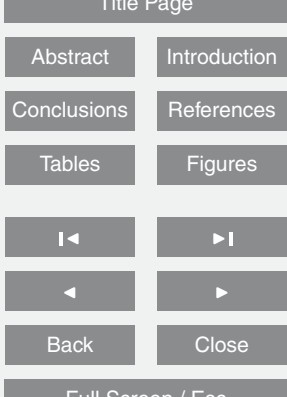



Taylor, J. R.: An Introduction to Error Analysis, 2nd edn., University Science Books, Sausalito, California, 327 pp., 1997.

Theule, J. I., Liébault, F., Loye, A., Laigle, D., and Jaboyedoff, M.: Sediment budget monitoring of debris-flow and bedload transport in the Manival Torrent, SE France, Nat. Hazards Earth Syst. Sci., 12, 731–749, doi:10.5194/nhess-12-731-2012, 2012.

Theule, J. I., Liébault, F., Laigle, D., Loye, A., and Jaboyedoff, M.: Channel scour and fill by debris flows and bedload transport, Geomorphology, 243, 92–105, 2015.

Van Dine, D. F.: Debris flow and debris torrents in the Southern Canadian Cordillera, Can. Geotech. J., 22, 44–68, 1985.

Van Steijn, H.: Debris-flow magnitude–frequency relationships for mountainous regions of Central and Northwest Europe, Geomorphology, 15, 259–273, 1996.

Veyrat-Charvillon, S.: Elaboration d'une méthode de prédiction du volume maximal d'une lave torrentielle (PREVENT), PhD thesis, Université Blaise Pascal, Clermont-Ferrant 2, 354 pp., 2005.

Veyrat-Charvillon, S. and Memier, M.: Stereophotogrammetry of archive data and topographic approaches to debris-flow torrent measurements: calculation of channel-sediment states and a partial sediment budget for Manival torrent (Isère, France), Earth Surf. Proc. Land., 31, 201–219, 2006.

Wu, Y. and Cheng, H.: Monitoring of gully erosion on the Loess Plateau of China using a global positioning system, Catena, 63, 154–166, 2005.

Zimmermann, M., Mani, P., and Romang, H.: Magnitude–frequency aspects of alpine debris flows, Eclogae Geol. Helv., 90, 415–420, 1997.

Discussion Paper | Discussion Paper | Discussion Paper | Discussion Paper |

**ESURFD**

doi:10.5194/esurf-2015-48

**Headwater sediment dynamics in debris flow catchment**

A. Loye et al.

# ESURFD

doi:10.5194/esurf-2015-48

**Headwater sediment dynamics in debris flow catchment**

A. Loye et al.

**Table 1.** TLS dates of acquisitions. Note that for the analysis, the 2nd survey was merged with the 1st one (see text for details).

| Monitoring period | Survey date | Period ID |
| --- | --- | --- |
| 1st | 1 Apr 2009–12 Jul 2009 | MP1 |
| 2nd | 12 Jul 2009–30 Aug 2009 | merged with MP1 |
| 3rd | 30 Aug 2009–11 Nov 2009 | MP2 |
| 4th | 11 Nov 2009–8 Jul 2010 | MP3 |

**ESURFD**

doi:10.5194/esurf-2015-48

**Headwater sediment dynamics in debris flow catchment**

A. Loye et al.

**Table 2.** TLS data and surface coverage characteristics of the 5 subcatchments from MP1. As the view points and parameters of acquiring remained similar, the values are essentially the same within all surveys.

| Subcatchment name | Surface[*] | | Lidar Data Survey | | | | Scanned area[*] | |
| --- | --- | --- | --- | --- | --- | --- | --- | --- |
| | Total [km$^2$] | Vegetated coverage [%] | Number of points | Mean spacing [m] | Mean range [m] | Mean density [pts m$^{-2}$] | Total [km$^2$] | non vegetated [%] |
| Col du Baure | 0.29 | 43.0 | 37 625 236 | 0.055 | 131 | 340 | 0.11 | 84 |
| Roche Ravine | 0.30 | 20.5 | 43 736 412 | 0.071 | 278 | 251 | 0.17 | 79 |
| Manival | 0.35 | 9.1 | 40 192 976 | 0.096 | 349 | 141 | 0.28 | 90 |
| Grosse Pierre | 0.08 | 9.0 | 9 703 449 | 0.110 | 447 | 145 | 0.07 | 97 |
| Genievre | 0.35 | 26.6 | 19 886 472 | 0.108 | 311 | 109 | 0.18 | 79 |
| Production Zone | 1.36 | 22.7 | 151 144 545 | 0.081 | 275 | 219 | 0.82 | 84 |

[*] Topographic surface area.

# ESURFD

doi:10.5194/esurf-2015-48

**Headwater sediment dynamics in debris flow catchment**

A. Loye et al.

**Table 3.** Registration and georeferencing standard deviation (in cm) of the position uncertainty on a point to point basis period that was used to derive the LoD at 95 % confidence interval and subsequently the detected minimum volume of geomorphic features.

| Sub-catchment | $2\sigma$ co-registered [cm] | | | | $2\sigma$ co-georeferencing (LoD) [cm] | | | $2\sigma$ Taylor uncertainty[*] [cm] $(\sigma_{d_{reg}} = \sqrt{\sigma_{d_{PC1}}^2 + \sigma_{d_{PC2}}^2})$ | | |
|---|---|---|---|---|---|---|---|---|---|---|
| | Survey | | | | Monitoring period | | | Monitoring period | | |
| | 1st | 2nd | 3rd | 4th | 1st | 2nd | 3rd | 1st | 2nd | 3rd |
| Col du Baure | 1.9 | 1.7 | 1.5 | 1.5 | 5.9 | 6.9 | 6.9 | 5.1 | 4.5 | 4.2 |
| Roche Ravine | 3.2 | 2.9 | 2.6 | 2.7 | 8.4 | 9.4 | 9.0 | 8.6 | 7.7 | 7.5 |
| Manival | 4.6 | 4.1 | 3.0 | 3.4 | 9.6 | 10.2 | 12.2 | 12.3 | 10.2 | 9.1 |
| Grosse Pierre | 4.1 | 3.0 | 3.3 | 3.3 | 10.6 | 10.6 | 12.2 | 10.2 | 8.9 | 9.3 |
| Genièvre | 3.7 | 3.6 | 3.2 | 3.6 | 6.7 | 7.6 | 8.3 | 10.3 | 9.6 | 9.6 |

[*] pc = point cloud used to generate the map (point cloud) of difference in 3-D.

**ESURFD**

doi:10.5194/esurf-2015-48

**Headwater sediment dynamics in debris flow catchment**

A. Loye et al.

**Table 4.** Sediment budget (in m$^3$) of the Manival torrent established after noticeable events using the morphological approach after Theule et al. (2012). The torrent recharge (sediment input) is estimated from in-storage changes in channels and volumes deposited in the sediment trap (output).

| Monitoring Period | | Survey dates in the torrent | Sediment Output | Storage Change | Channel Erosion | Channel Deposition Input | Sediment Input | Total sediment |
|---|---|---|---|---|---|---|---|---|
| 1st | #1 | 6 Jul 2009–28 Aug 2009 | 1873 ± 62 | −2034 ± 559 | 5232 ± 136 | 3199 ± 63 | 0–63 | 0–63 |
| 2nd | #2 | 30 Aug 2009–7 Oct 2009 | 0 | 789 ± 84 | 1409 ± 31 | 2197 ± 53 | 736–842 | 934–1102 |
| | #3 | 8 Oct 2009–12 Nov 2009 | 302 ± 36 | −73 ± 66 | 1546 ± 36 | 1473 ± 31 | 198–260 | |
| 3rd | #4 | 13 Nov 2009–1 Jun 2010 | 580 ± 45 | −580 ± 81 | 1961 ± 45 | 1372 ± 36 | 0–36 | 174–844[*] |
| | #5 | 2 Jun 2010–8 Jun 2010 | 3320 ± 176 | −3052 ± 272 | 7658 ± 178 | 4606 ± 93 | 0–537 | |
| | #6 | 9 Jun 2010–8 Oct 2010 | 819 ± 46 | −608 ± 82 | 2246 ± 46 | 1637 ± 36 | 174–246 | |

[*] The TLS survey MP3 lasted until 8 July 2010; #6 were not considered for the analysis of the sediment budgets.

**Table 5.** Overall headwater sediment budget recorded during the three survey periods and net sediment balance of the 16 months of monitoring. Sediment budgets for each catchment subsystem are detailed in the Supplement.

| 1st monitoring | Volume Total [m³] | | | | | |
|---|---|---|---|---|---|---|
| period | Hillside | | Channel | | Headwater | |
| Rockfall | 99.4 | ±5.9 | | | 99.4 | ±5.9 |
| Deposition | 408.2 | ±35.4 | 149.2 | ±10.9 | 557.4 | ±46.3 |
| Erosion | 547.2 | ±49.5 | 636.4 | ±43.3 | 1183.5 | ±92.8 |
| Subtotal | −238.3 | ±61.2 | −487.2 | ±44.7 | −725.6 | ±103.9 |
| 2nd monitoring | Volume Total [m³] | | | | | |
| period | Hillside | | Channel | | Headwater | |
| Rockfall | 50.5 | ±3.0 | | | 50.5 | ±3.0 |
| Deposition | 181.8 | ±12.2 | 127.2 | ±8.0 | 309.0 | ±20.5 |
| Erosion | 639.8 | ±27.1 | 522.5 | ±19.4 | 1162.3 | ±46.4 |
| Subtotal | −508.5 | ±29.9 | −395.3 | ±23.4 | −903.7 | ±50.9 |
| 3rd monitoring | Volume Total [m³] | | | | | |
| period | Hillside | | Channel | | Headwater | |
| Rockfall | 3424.9 | ±89.1 | | | 3424.9 | ±21.4 |
| Deposition | 3163.5 | ±147.9 | 1105.5 | ±36.4 | 4269.0 | ±175.6 |
| Erosion | 1941.6 | ±72.8 | 650.8 | ±28.8 | 2592.4 | ±91.6 |
| Subtotal | −2203.0 | ±187.4 | 454.7 | ±46.5 | −1748.3 | ±199.2 |
| Total | Volume Total [m³] | | | | | |
| monitoring | Hillside | | Channel | | Total | |
| Rockfall | 3574.7 | ±97.9 | | | 3574.7 | ±30.3 |
| Deposition | 3753.5 | ±195.6 | 1381.9 | ±55.6 | 5135.4 | ±251.3 |
| Erosion | 3128.5 | ±149.4 | 1809.7 | ±91.3 | 4938.2 | ±240.8 |
| Subtotal | −2949.8 | ±264.9 | −427.8 | ±106.9 | −3377.6 | ±361.4 |

Discussion Paper | Discussion Paper | Discussion Paper | Discussion Paper |

**ESURFD**

doi:10.5194/esurf-2015-48

**Headwater sediment dynamics in debris flow catchment**

A. Loye et al.

Discussion Paper | Discussion Paper | Discussion Paper | Discussion Paper |

**ESURFD**

doi:10.5194/esurf-2015-48

**Headwater sediment dynamics in debris flow catchment**

A. Loye et al.

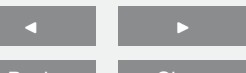

**Table 6.** Rock slope debris production rate estimated from the inventory analysis using power law distribution of volume for potential rockfall (Fig. 10).

| Class of volume in m$^3$ | $10^{-3}$–$10^{-2}$ | $10^{-2}$–$10^{-1}$ | $10^{-1}$–1 | 1–10 | $10^1$–$10^2$ | $10^2$–$10^3$ | $10^3$–$10^4$ | $10^4$–$10^5$ | $10^5$–$10^6$ | $10^6$–$10^7$ |
|---|---|---|---|---|---|---|---|---|---|---|
| Measured frequency (per year) | 143 (112.5) | 742 (583.7) | 789 (620.7) | 168 (132.2) | 19 (14.95) | 3 (2.36) | 1 (0.79) | | | |
| Calculated frequency | 36990 ±4366 | 5621 ±581 | 854 ±86 | 130 ±9.6 | 19.7 ±1.2 | 3.0 ±0.14 | 0.46 ±0.015 | 0.069 ±0.0013 | 0.011 ±1 × 10$^{-4}$ | 0.0016 ±1.2 × 10$^{-5}$ |
| Cumulative Measured Frequency | 1467 | 1355 | 772 | 152 | 19 | 3.1 | 0.79 | | | |
| Cumulative Calculated Frequency | 43 619 ±5043 | 6629 ±677 | 1007 ±97 | 153 ±11 | 23 ±1.58 | 3.5 ±0.198 | 0.54 ±0.018 | 0.08 ±0.0014 | 0.01 ±1.1 × 10$^{-4}$ | 0.0016 ±1.2 × 10$^{-5}$ |
| Fallen volume per year [m$^3$] | 102 ±12 | 155 ±16 | 236 ±19 | 358 ±26 | 544 ±32 | 827 ±37 | 1257 ±39 | 1911 ±32 | 2904 ±8 | 4413 ±51 |
| Total fallen volume per year [m$^3$] | 298 ±43 | 454 ±59 | 689 ±79 | 1047 ±105 | 1592 ±136 | 2419 ±172 | 3676 ±210 | 5587 ±241 | 8491 ±249 | 12 903 ±305 |
| Cliff area | 826 804 m$^2$ (only the topographic rock slope surface) | | | | | | | | | |
| Erosion rate [mm] | 0.36 ±0.05 | 0.54 ±0.07 | 0.83 ±0.1 | 1.3 ±0.1 | 1.9 ±0.2 | 2.9 ±0.2 | 4 ±0.3 | 6.8 ±0.3 | 10.2 ±0.3 | 15.6 ±0.4 |

## ESURFD

doi:10.5194/esurf-2015-48

**Headwater sediment dynamics in debris flow catchment**

A. Loye et al.



**Figure 1.** (Inset) Map of the study area; the Manival catchment is displayed in full red and the impressive debris fan is streaked. (Outset) Aerial view of the Manival catchment; sediment supply concentrates exclusively in the headwater (production zone) as erosion activity from the middle and lower catchment is not connected to the torrent (zone of transfer).

## ESURFD

doi:10.5194/esurf-2015-48

**Headwater sediment dynamics in debris flow catchment**

A. Loye et al.

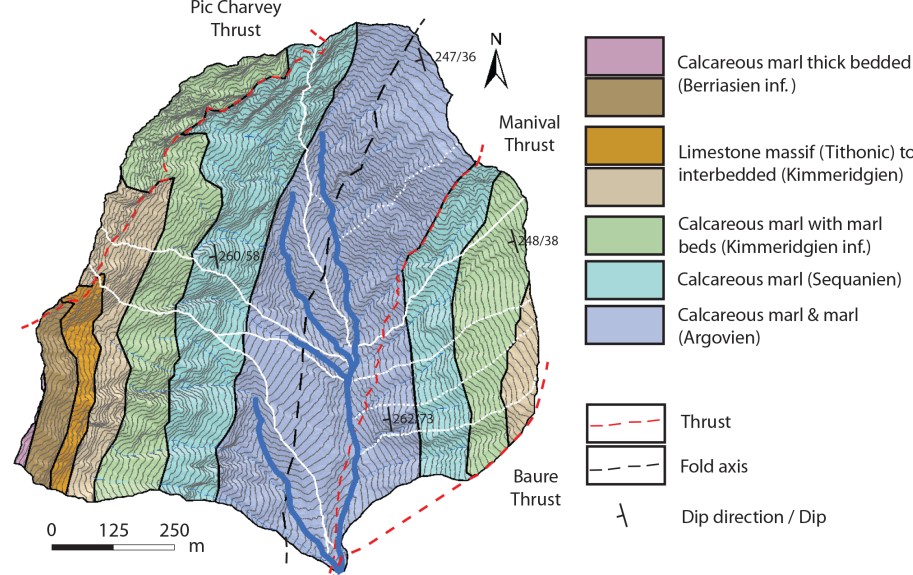

**Figure 2.** Geological map of the catchment headwater, after Gidon (1991).

# ESURFD

doi:10.5194/esurf-2015-48

**Headwater sediment dynamics in debris flow catchment**

A. Loye et al.

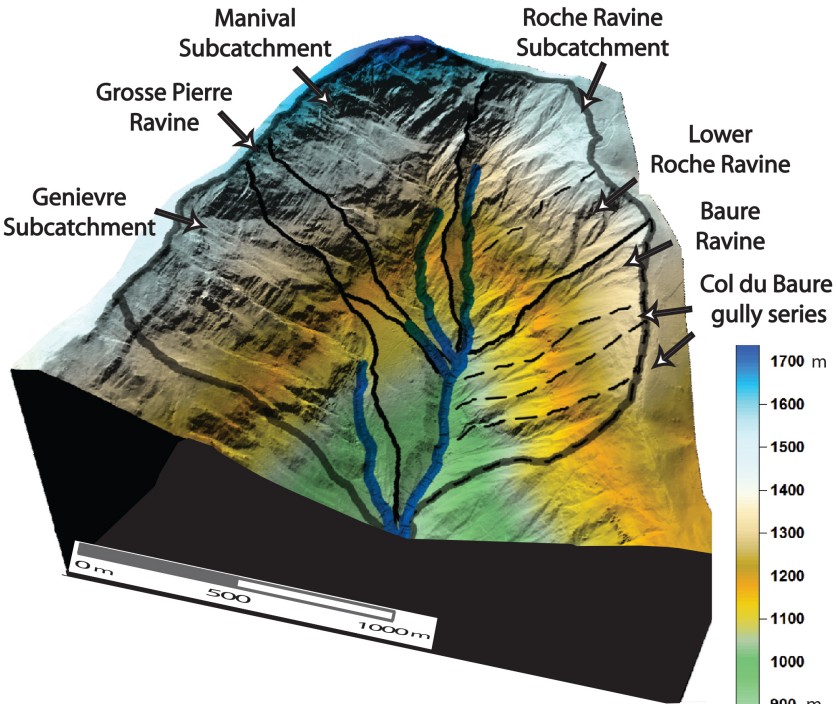

**Figure 3.** 3-D shaded view of the Manival headwater (production zone of Fig. 1), showing the first-order debris flow channels and their respective contributing area. For the ease of analysis, the Roche Ravine and Col du Baure subcatchments in the east side were further subdivided according to their gully complex. The topography of the production zone consists of a narrowly-confined valley head, delimited side west by series of rock walls and scree-mantled deposits separated by rock couloirs, and by steep rock and talus slopes divided by gullies side east (see also Fig. 4 for more details on sediment processes).

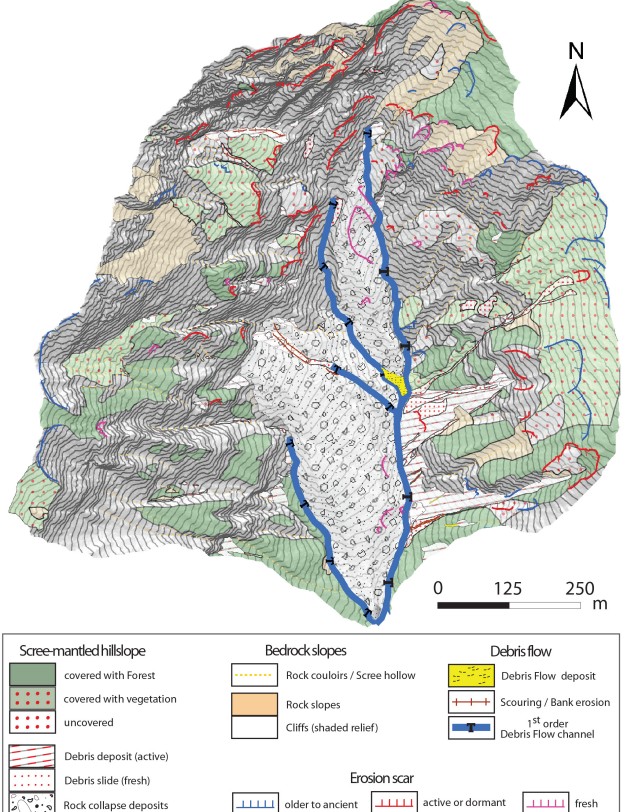

**Figure 4.** Geomorphic process map (contour interval: 20 m) illustrating the spatial pattern of sediment sources and transfer in the first-order channel complex. Note the impressive rock collapse deposits now crossed by four first-order debris channels. Their bed incision is strongly constrained by series of check dams, but erosion scars all along the deposit suggest that the reaches are still subject to lateral erosion.

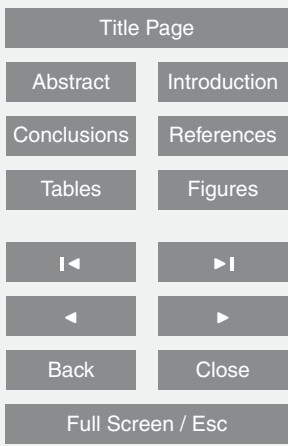

Discussion Paper | Discussion Paper | Discussion Paper | Discussion Paper

**ESURFD**

doi:10.5194/esurf-2015-48

**Headwater sediment dynamics in debris flow catchment**

A. Loye et al.

Discussion Paper | Discussion Paper | Discussion Paper | Discussion Paper

# ESURFD

doi:10.5194/esurf-2015-48

**Headwater sediment dynamics in debris flow catchment**

A. Loye et al.

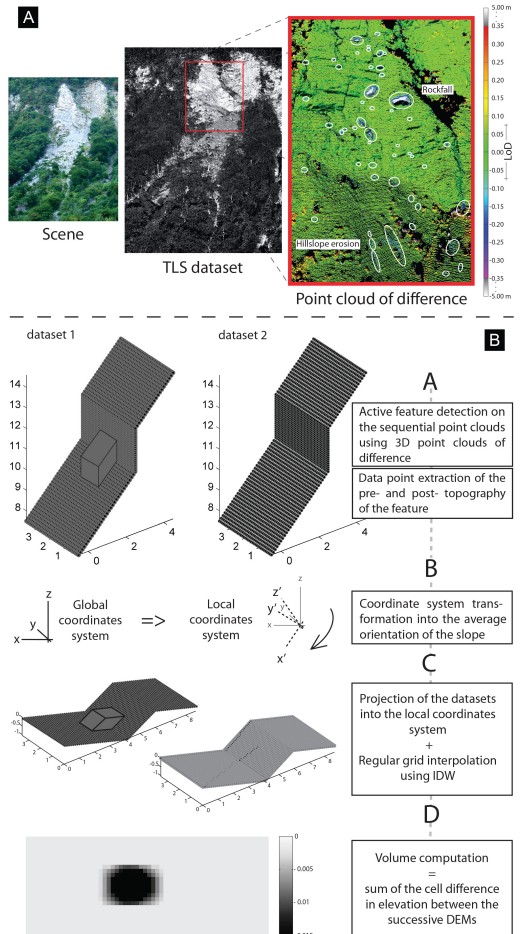
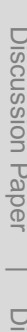

**Figure 5. (a)** 3-D detection and **(b)** extraction and volume computation method of an individual active feature provided by two successive point cloud datasets.

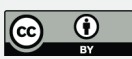

**ESURFD**

doi:10.5194/esurf-2015-48

**Headwater sediment dynamics in debris flow catchment**

A. Loye et al.

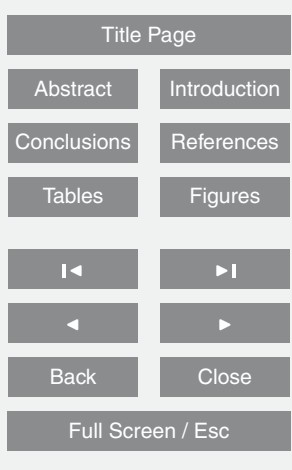

| Histogram Statistics ICP : | | Georeferencing Point Cloud April 2009 - July 2010 |
|---|---|---|
| Subcatch. name | Manival | |
| Mean ( μ ) | 0.000104 | |
| Std Dev. ( σ ) | 0.060718 | |
| Minimum (min) | -0.212658 | |
| Maximum (max) | 0.212658 | |
| Peak | 48593 | |
| Number of Points | 3526850 | |

**Figure 6.** Distribution of the distance between two survey point clouds after the process of georeferencing using ICP procedure. The distance approaches normal distribution with a zero mean, showing that errors generated by multiple scan registration and point cloud survey georeferencing are Gaussian, random and independent. Data are given in meters.

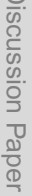

# ESURFD

doi:10.5194/esurf-2015-48

## Headwater sediment dynamics in debris flow catchment

A. Loye et al.

1<sup>st</sup> Monitoring Period
April 2009 - September 2009

**Rockslope Erosion**
**Volume (m³)**
- 0.00 - 0.25
- 0.25 - 0.75
- 0.75 - 5.0
- 5.0 - 23
- 23 - 25

**Deposition**
**Volume (m³)**
- 0.02 - 2.0
- 2.0 - 5.0
- 5.0 - 15
- 15 - 60
- 60 - 270

**Hillslope Erosion**
**Volume (m³)**
- 0.00 - 5
- 5 - 20
- 20 - 50
- 50 - 200
- 200 - 220

**No LiDAR Coverage**

N

0 | 125 | 250
Meters

**Figure 7.** Geomorphic activity revealed by comparing the topographic differences of the two successive TLS surveys operated in April and August 2009. The sediment budgets are detailed for each subcatchment in Fig. 13.

Discussion Paper | Discussion Paper | Discussion Paper | Discussion Paper

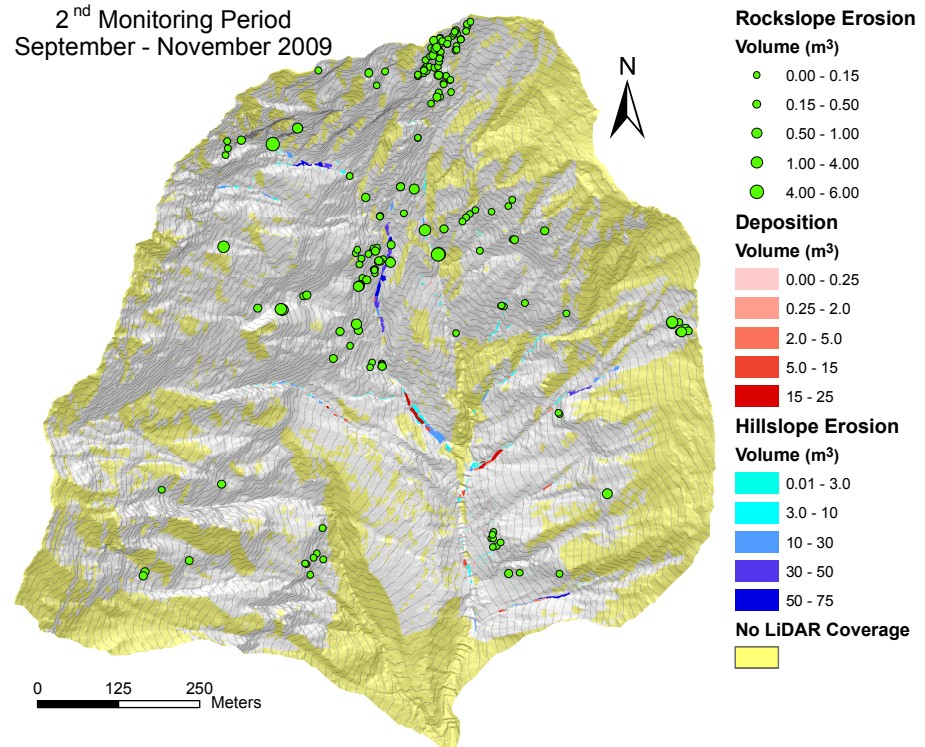

**Figure 8.** Geomorphic activity revealed by comparing the topographic differences of the two successive TLS surveys operated in August and November 2009. The sediment budgets are detailed for each subcatchment in Fig. 14.

**ESURFD**

doi:10.5194/esurf-2015-48

**Headwater sediment dynamics in debris flow catchment**

A. Loye et al.

Discussion Paper | Discussion Paper | Discussion Paper | Discussion Paper

ESURFD

doi:10.5194/esurf-2015-48

A. Loye et al.

3 rd Monitoring Period
November 2009 - July 2010

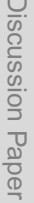

N

**Rockslope Erosion**
Volume (m³)
- 0.00 - 10
- 10 - 60
- 60 - 300
- 300 - 700
- 700 - 1100

**Deposition**
Volume (m³)
- 0.09 - 25
- 25 - 100
- 100 - 200
- 200 - 600
- 600 - 1200

**Hillslope Erosion**
Volume (m³)
- 0 - 5
- 5 - 15
- 15 - 60
- 60 - 180
- 180 - 270

**No LiDAR Coverage**

0   125   250   Meters

**Figure 9.** Geomorphic activity revealed by comparing the topographic differences of the two successive TLS surveys operated in November 2009 and July 2010. The sediment budgets are detailed for each subcatchment in Fig. 15.

Discussion Paper | Discussion Paper | Discussion Paper | Discussion Paper

**ESURFD**

doi:10.5194/esurf-2015-48

**Headwater sediment dynamics in debris flow catchment**

A. Loye et al.



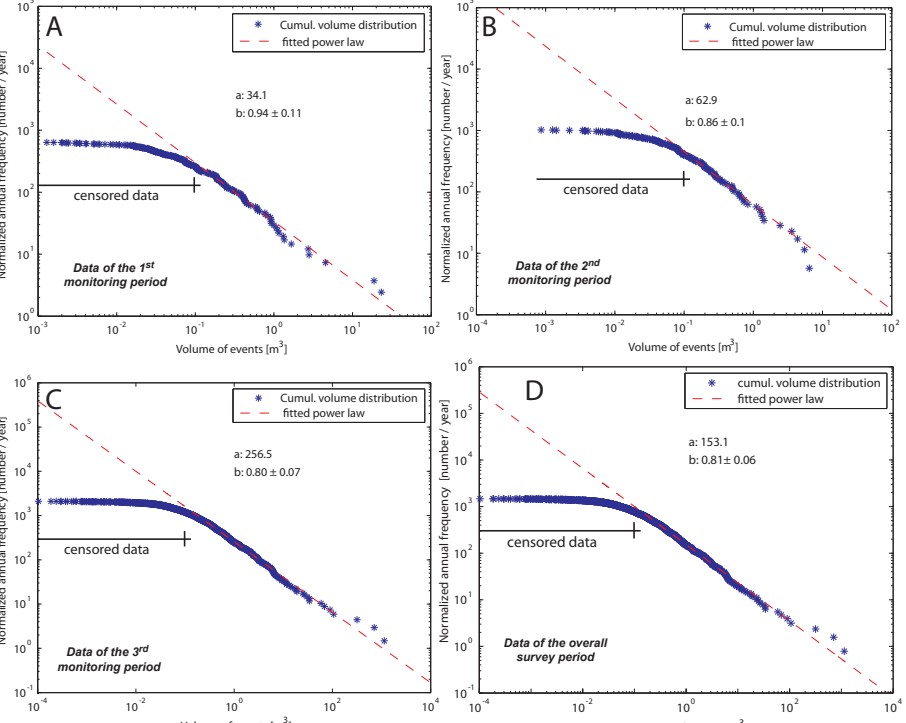

**Figure 10.** Cumulative volume distribution of the rockfall observed during the first **(a)**, the second **(b)**, the third monitoring period **(c)** and over the entire study time of 16 months **(d)**. For each dataset, the power law is fitted for volumes larger than 0.1 m$^3$. Below this threshold volume, the distribution exhibits a roll-over that progressively reaches a quasi-constant frequency for the smallest detected volumes.

**ESURFD**

doi:10.5194/esurf-2015-48

**Headwater sediment dynamics in debris flow catchment**

A. Loye et al.

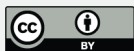

**Figure 11.** Torrent in-channel storage changes per unit length and sediment budgets of cumulative volumes transported in the torrent from the headwater outlet to the sediment trap downstream for each monitoring period (MP). The torrent recharge (sediment input) was estimated given the in-storage change and the volume deposited in the sediment trap (see Table 4 for details on values) (modified from Theule et al., 2012).

Discussion Paper | Discussion Paper | Discussion Paper | Discussion Paper |

# ESURFD

doi:10.5194/esurf-2015-48

**Headwater sediment dynamics in debris flow catchment**

A. Loye et al.

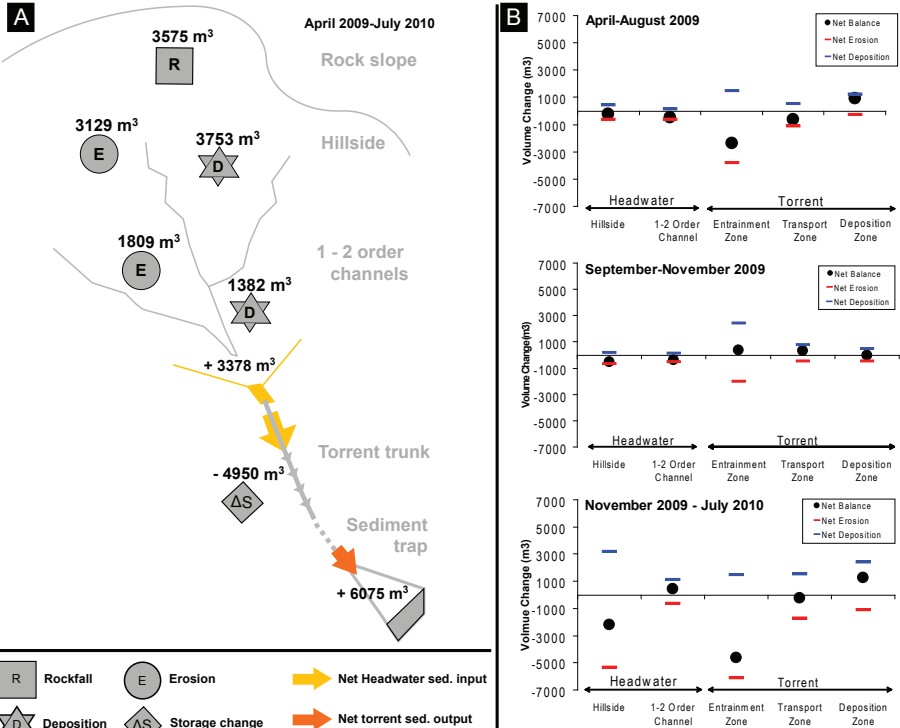



**Figure 12. (a)** Global sediment budget and **(b)** net sediment balance for each monitoring period showing the overall transfer dynamic from debris source zone in the headwater to the apex of the fan through the torrent observed during the period of investigation.

## ESURFD

doi:10.5194/esurf-2015-48

**Headwater sediment dynamics in debris flow catchment**

A. Loye et al.

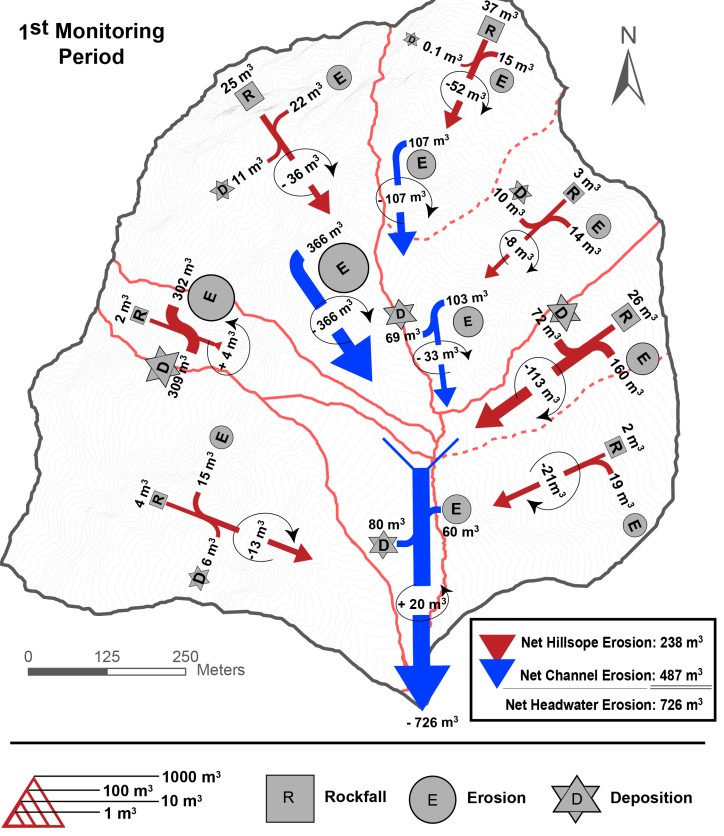

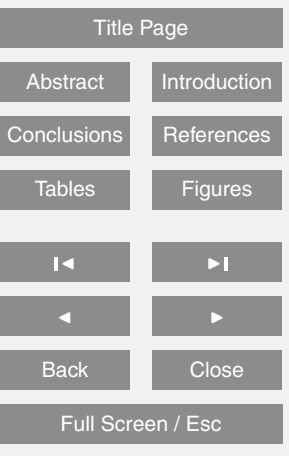

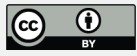

**Figure 13.** Overall headwater sediment budget observed during the 1st monitoring period revealing the sediment dynamic through the spring-summer season and the net balance of sediment recharge in the downstream torrent for the several months preceding the august 2009 debris flow.

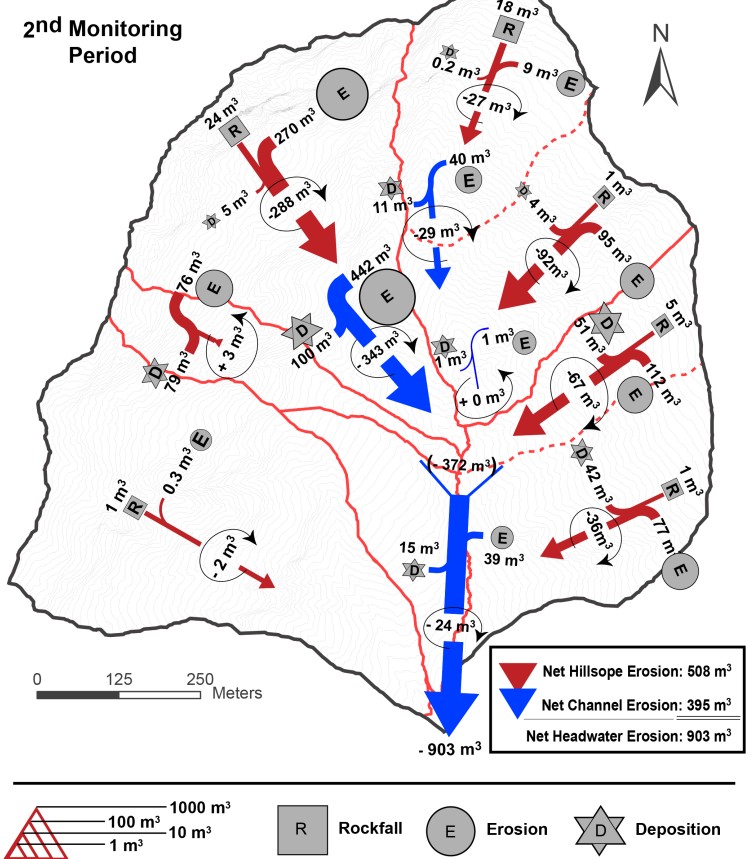

**Figure 14.** Overall headwater sediment budget observed during the 2nd monitoring period revealing the sediment dynamic and the net balance of sediment recharge in the downstream torrent during the autumn.

ESURFD

doi:10.5194/esurf-2015-48

Headwater sediment dynamics in debris flow catchment

A. Loye et al.

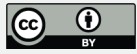

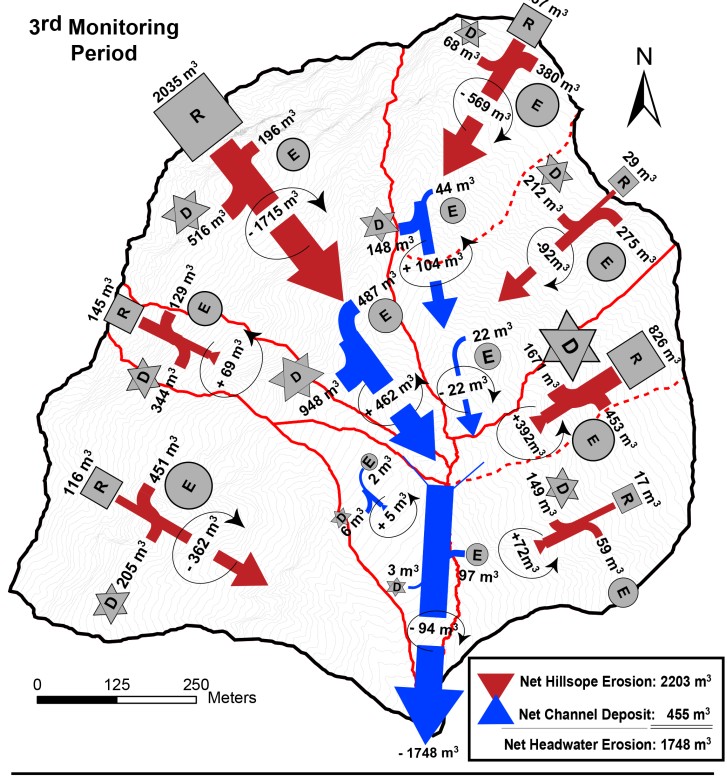

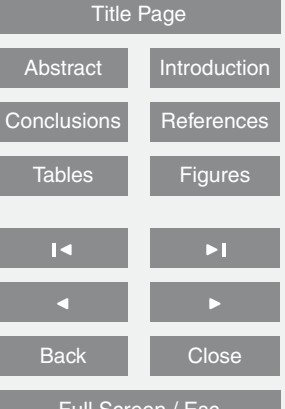

# ESURFD

doi:10.5194/esurf-2015-48

**Headwater sediment dynamics in debris flow catchment**

A. Loye et al.

**Figure 15.** Overall headwater sediment budget observed during the 3rd monitoring period revealing the sediment dynamic through the winter–spring and the net balance of sediment recharge in the downstream torrent for the period preceding the June 2010 debris flow.

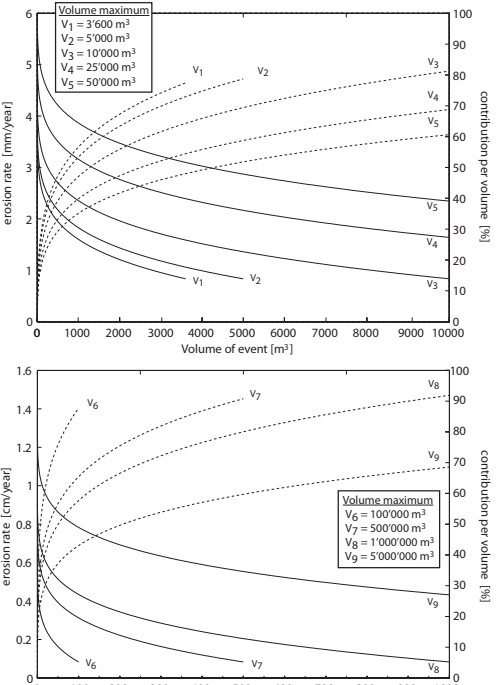

**Figure 16.** (continuous line) Erosion rate as function of size of events for a certain volume of production (potential maximum volume $V_{1...9}$), considering that rockfall volume distribution observed at Manival follows power law behaviour (Table 4). (dash line) Contribution of each class of volumes to the erosion rate showing the significant effect of large slope failures. For a maximum volume eroded of $3600\,\mathrm{m}^3\,\mathrm{yr}^{-1}$ ($V_1$), the $1000\,\mathrm{m}^3$ rockfall event contributes for 60 %, while events less than $100\,\mathrm{m}^3$ induce less than 20 % of erosion, although of much higher frequency; a $100\,000\,\mathrm{m}^3$ rockslide would generate 70 % of a total of material eroded of $500\,000\,\mathrm{m}^3$ ($V_7$) over a century.

Discussion Paper | Discussion Paper | Discussion Paper | Discussion Paper |

**ESURFD**

doi:10.5194/esurf-2015-48

**Headwater sediment dynamics in debris flow catchment**

A. Loye et al.