# Peer review of "HEADWATER SEDIMENT DYNAMICS IN DEBRIS FLOW"

_Earth Surface Dynamics, 2015_

## Referee Comment (RC1) · Anonymous Referee #1 · 11 Feb 2016

This is an interesting paper. The authors presented an analysis on the headwater sediment dynamics in a debris flow catchment located in the French Alps. The catchment was surveyed periodically during 16 months using terrestrial laser scanning (TLS) with the purpose to analyze the coupling between sediment dynamics and torrent responses in terms of debris flow events. This is one of the first papers I've seen where the TLS was used so extensively on a landscape for the analysis of Earth surface processes at catchment scale. It absolutely deserves to be published in ESurf, and probably it will be a solid reference point for the Earth science community in the future. Having said that, the general impression is that the paper is a little long, not so easy to read. In my opinion it should be simplified in several sections, summarizing the re-

sults and making the discussion simple and direct. The conclusion section needs to be significantly reduced, while adding one or two sentences on future challenges of such kind of analysis.

Minor issues

The equations should be numbered as commonly happens in scientific papers. Green circles in Fig. 9 should present the same style of Fig. 8 and 7, so with a black rings. I suggest to add 2-3 figures representing detailed views of erosional areas, in addition to one picture showing the TLS during the survey.

Overall I recommend moderate-to-major changes, very easy to address.

---

## Referee Comment (RC2) · O. Sass (Referee) · 15 Feb 2016

This is a very interesting paper on sediment transport and sediment budget in an alpine catchment. The investigation was thoroughly performed and yielded an impressive amount of data. The data were carefully evaluated including an uncertainty analysis and are presented in (mostly) very good figures. The paper definitively warrants publication in ESurf after minor to moderate revisions.

The overall concept of the paper is very good. However, there are some problems with the presentation. The three most important points are:

The paper is not exactly easy to read because of the many details which are not clearly
laid out. I understand that this is a huge body of data; and as methods, errors etc should be treated in sufficient depth, there is not very much potential for cuts. However, this little potential should be used.

The F/M-approach for rockfalls is very good. Nevertheless, it is not strictly necessary in the context of the contemporary sediment budget. I would suggest to leave it out and discuss this in detail in a second paper. This includes the section in the Methodology, but also in the Discussion, e.g. P19 L18-21; P20 L1-10; and Fig. 16.

The English is generally good but there are some (mostly minor) problems which a native speaker could easily correct to improve readability. (One is the rather odd, persistently wrong use of the word "any").

The Abstract should summarize aims, study sites, methods, results and the most important conclusions. Currently, it starts with a weird sentence; some facts are missing (time period of investigation, basic information on the catchment); and the results are not well structured.

In the Study Area section, any information on climate is missing. Quite frequently throughout the paper the authors mention e.g. "response to heavy rainstorms that occur regularly" (P5 L5). What is a "heavy rainstorm" in the area? Information on mean annual precipitation, rainstorm intensities, MAAT is entirely missing. Further example P16 L9: "series of rainfall of low to moderate intensity" – it is never defined what low intensity is.

The Introduction is well-written with a good number of references – nothing to improve on.

There is too much repetition and re-consideration of the own data in the Discussion - towards the end, the paper gets tiresome to read. Consider to shorten the Discussion, with a stronger focus on the comparison to other work and on the geomorphic implications. More literature would be good in some places (e.g. P20 L14/15, and the entire

**ESurfD**
sections 6.2 and 6.3). If other work is cited for comparison, it is mostly from French collegues working in the area. Most of paragraph 6.5 is rather speculative and could be left out (e.g. first sentences, and L11 ff)

The Conclusion is the weakest section of the paper. Here, there would be the chance to summarize the multitude of details in a clear way, maybe using bullet points. However, the paragraph is as confusing as much of the Results and Discussion are. By the start of the Conclusions, the reader is desperately waiting for some clear, summarizing facts: Which are the key findings after such a lot of details? Furthermore, the Conclusions should summerize the already presented findings and NOT come up with new ideas (e.g. lines 5-7 are unclear and something new; wet snow avalanches (L16) have never been mentioned before, and so on). Quite a lot of the Conclusions (if not all) is in fact Discussion (e.g. P26 L1-4).

**Figures/Tables**

Table 1. TLS dates of acquisitions: In the text you speak of a "five-day survey" while here it seems to be one defined day for each survey. Do you mean "five surveys of one day each" in the text body? (If so, it seems like an impressive survey speed to achieve 50 scans from 9 viewpoints in a day..?)

Table 4. Sediment budget: It is not clear to me what the difference between "channel deposition input" and "sediment input" is; and what "total sediment" means

Figure 1: The inset is not ideal as you can't see any topography at all, and a legend of the geological units is missing.

Figure 2: I assume that the thick blue lines are the main torrents. However, the white lines are not explained in the legend.

Figure 3: The Figure does not provide much important information and should be left out. I think that Fig. 1 and Fig. 4 are sufficient; maybe some of the names could be added in Fig. 1.

**ESurfD**
Fig. 4: Are the black symbols check dams?

Fig. 7,8,9: It would be better to use the same colour charts throughout the three consecutive figures. Fig. 9 should use the same legend as in Fig. 7 and 8 (black circle around the rockslope erosion dots)

Fig. 10: It makes not much sense to show the data for each period - the entire study time (d) is sufficient.

Figure 12: Very good figure - after some trying to understand, all important parts of the budget become clear.

Fig. 13,14,15 are excellent figures.

Fig. 16: leave out

Minor comments

P5 L8: what does "any" mean in this context?

P5 L9: delete "a"

P5 L9: "old rock deposits flooring the hillslope side west (Fig. 3)" - This can't be seen in Fig. 3 (more in Fig. 4). I suggest to leave out Fig. 3.

P5 L11: resulting in

P5 L13: five (spell out numbers

P8 L1: computes

- P8 L4: what does "of similar sign" mean?
- P9 L13: "cell differences lying below a given threshold" which was this threshold?
- P12 L19: "did not show any" instead of "showed any"
- P13 L15: spatial extent
- P13 L18: did not show any
- P13 L22: signs
- P13 L23: rilling areas
- P14 L10: rock slope erosion
- P14 L20: rockfalls
- P15 L13: "almost without any" instead of " with almost any"
- P16 L5: in the torrent
- P16 L9: "no" instead of "any"
- P16 L14: rainfalls
- P16 L19: "no" instead of "any"
- P16 L20: "No" instead of "Any"
- P17 L1: Synthesis
- P17 L8: "of which" instead of "whereas"
- P17 L14 "acts as recharging the torrent" reword, what does this mean?
- P18 L15: delete "degradation"
- P18 L21: "the bedrock surface is often highly fractured, suggesting frost shattering."

**ESurfD**
- This is a case of wrong reasoning and self-fulfilling prophecy (which is frequently found in this context). The fact that the bedrock surface is highly fractured is, it itself, no indication of frost action. It might be fractured for tectonic-lithological reasons, or by the impact of any other type of weathering (hydration, solution processes, wetting/drying...)

P20 L3: "No" instead of "Any"

P20 L27: "For the entire area" instead of "globally" (which would be earth-wide)

- P21 L21: "sediment density" instead of "process density"?
- P23 L7: exhaustion instead of starvation

P23 L8: "no" instead of "any"

**ESurfD**

---

## Referee Comment (RC3) · Anonymous Referee #3 · 23 Feb 2016

The paper entitled "headwater sediment dynamics in debris flow catchment: implication of debris supply using high resolution topographic surveys" presented by Loye et al. constitutes a very interesting study on debris supply rythms for debris flow initiation. The paper is long, but presents lots of results, also using numerous previous studies that have been carried out in the same torrent. The paper would gain being presented in a more straightforward way, and would benefit of an English editing. Here are some comments the authors might consider while revising their manuscript:

P. 2, line 25: could you specify how you consider snowmelt identical to rainfall? Snowmelt has been proven as a different triggering factor from rainfall (intense or long lasting) (ex. Decaulne et al., 2005 in Geografiska Annaler), and has also been denied,

or with a very secondary action in some studies (ex Jomelli 2004 in Climate Change). P. 3, line 3-4: how do you compare debris flow amounts with initiation mechanisms in a quantitative way? P. 6, line 27: what are the criteria to recognize and then eliminate erroneous points? P. 6, line 17-20: has this be previously validated by experiments? Please use a reference. P. 10: equations should be numbered. P. 12, line 4 'whereas b tends to be rather site independent'; why don't you say 'is site independent'? why are you so cautious? P. 12, line 16: what make you select the duration of each period, as they are uneven in time. Please be more specific on this and provide an explanation earlier in the pape. P. 18, line 18: you specify that freeze-thaw is a major key for debris supply, contributing to rockfall that will later feed the debris flows. You may also consider the triggering factors for debris flow initiation, and especially snowmelt of snow avalanche deposits (Bardou and Delaloye, 2004 in NHESS). P. 20, line 8: you mention 'about' before the calculation; that's a quite relevant word, which might lead to very different result according to the mentionned quantities.

Overall, this is a very significant contribution that will be abundantly cited in the future.

**ESurfD**

---

## Author Comment (AC1) · 4 Apr 2016

As all modifications done in the submitted manuscript were not completely described in the authors response, people are therefore invited to get to the chapter or paragraph of the revisited version of the manuscript that is related to the comments.

Please also note the supplement to this comment:
http://www.earth-surf-dynam-discuss.net/esurf-2015-48/esurf-2015-48-AC1-supplement.pdf

---

## Author Response (AR1)

Thank you very much for all comments and relevant suggestions. And « congrats » for the referee that gave his identity, which makes the discussion much more interesting when we know who makes comments.

Please find here a general response to the most critical comments and further below our answer to each comment one.

**Author's general response to the most important comments :**

Most of the general comments were directed toward the way the discussion and conclusion were written, underlining a need to simplify and clarify these two chapters in order to :

- set the most relevant outputs in a clear and short way
- bring out the key findings of this case study
- be more direct with statement by considering shorten both chapters
- get a better readability.

Hence, a great effort was put to review these two chapters, rewritting many parts of paragraphs with the following things in mind :

- get rid of the many details that could confuse the reader and get the paper hard to read
- keep only the general outputs and findings
- simplify the structure of the analysis (first part of the discussion) with only three subchapters, such as :
  - 1. debris supply from rockslope;
  - 2. debris supply from the hillside
  - 3. debris supply in the torrent
- avoiding diffuse analysis, this kind of statement that are split into 2 to 3 differents subchapters of the discussion
- avoiding recalling in a summarizing way the observation and facts described in the result part
- grouping both subchapter on the discussion of the analysis (last two subchapters of the discussion) to conduct the reader in a direct and clear matter to our key findings.

As these two last chapters underwent many changes, rephrasing and that the most relevant information of chapter 6.5 was dispatched in the other subchapters, all modifications done in the submitted manuscript are not completely described. The referees are therefore invited to get to the chapter or paragraph related to the comment.

Anonymous Referee #1 Received and published: 11 February 2016

Comments from Referees : the general impression is that the paper is a little long, not so easy to read. In my opinion it should be simplified in several sections, summarizing the results and making the discussion simple and direct. Author's response :

The paper was substantially reduced (more than 1 page) and many clarifications were brought out to improve readability (see general response above). The chap. 6.5 and figure 3 was left out. The discussion and conclusion was partly re-written in a more simple and direct way.

See as well our answers to referee #2 below.

**Author's changes in manuscript :**

Please, get to the chapters related to the comment to have the whole picture of the many modifications made.

Comments from Referees: The conclusion section needs to be significantly reduced, while adding one or two sentences on future challenges of such kind of analysis.

**Author's response :**

The conclusion was reduced. See as well our answers to referee #2 below.

The last sentences of the paper address the future challenges of such a kind of analysis directly related to this work. As mentioned in the paper, comparing this study with other close mountain catchments, and particularly in terms of headwater sediment regime, represents a challenge as well as very little literature exists (see P26/L18-19). Other sentences in the conclusion set the future challenges of this kind of studies (e.g. P26/L6)

Other challenges, such as improving TLS techniques (e.g. Schürch et al., 2012\*) or the development of more efficient identification and extraction procedure for TLS data processing (cost-effective tools for analyzing TLS data) are implicit in this paper as described in the methodology and need not to be further mentioned. As well consideration of the headwater – torrent sediment dynamics in terms of geomorphological and geological local setting as well as hydrometeorological solicitations are first implicit and then beyond the scope of the conclusion. This would make not really sense to add such a sentence at the end of the paper, when geological and morphological aspect of the catchment was not the point of this paper. For that, other papers about the Manival exist (see authors publication for exemple).

\*Schürch et al., 2012. Detection of surface change in complex topography using terrestrial laser scanning: application to the Illgraben debris-flow channel, ESPL.

**Author's changes in manuscript :**

For the chapter conclusion, see changes made after the comments from referee #2 below and read the chapter.

For the future challenges, no add !

Comments from Referees : The equations should be numbered as commonly happens in scientific paper

**Author's response :**

See same comment from referee #2 below.

Comments from Referees: Green circles in Fig. 9 should present the same style of Fig. 8 and 7, so with a black ring

**Author's response :**

See same comment from referee #2 below.

Comments from Referees : I suggest to add 2-3 figures representing detailed views of erosional areas, in addition to one picture showing the TLS during the survey.

**Author's response :**

Figure 4 provides a geomorphic process map and Figure 5 that illustrates the methodology provides a detailed view of an erosional area on the hillside of the Manival catchment. More illustration of the catchment and TLS survey can be found in Theule et al., 2012/2015 ; Loye et al., 2012 and in the PhD manuscript of both first and last author. As the paper is already long enough and TLS technique has become a common tool, the opinion of the authors is that there is no need for more figures.

**Author's changes in manuscript : No add !**

Referee #2: Oliver Sass oliver.sass@uni-graz.at Received and published: 15 February 2016

Thank you very much for providing your identity. Please find below our answers to your very constructive comments.

Comments from Referees: The paper is not exactly easy to read because of the many details which are not clearly laid out. I understand that this is a huge body of data; and as methods, errors etc should be treated in sufficient depth, there is not very much potential for cuts. However, this little potential should be used.

**Author's response :**

All referees comments were deeply considered and we did our best to improve the readability of the manuscript, especially by a deep reworking of discussion and conclusiont.

**Author's changes in manuscript :**

Cf. all referees comments !

Comments from Referees: The F/M-approach for rockfalls is very good. Nevertheless, it is not strictly necessary in the context of the contemporary sediment budget. I would suggest to leave it out and discuss this in detail in a second paper. This includes the section in the Methodology, but also in the Discussion, e.g. P19 L18-21; P20 L1-10; and Fig. 16.

**Author's response :**

We kept the F/M-approach in this paper, as this provides an original and comprehensive analysis of sediment production and recharge rates of headwaters when they are highly controlled by bedrock hillslope instabilities, like in our case. This kind of approach is part of contemporary sediment budget, since this can readily improve knowledge on sediment production and hence recharge threshold leading to debris flow, so their prediction, as we said in the conclusion.

But your suggestion could make sense if we would have the possibility to work more on that , which this is not the case unfortunatelly. Then we would write another paper, and link both paper together.

**Author's changes in manuscript :**

None !

Comments from Referees: The English is generally good but there are some (mostly minor) problems which a native speaker could easily correct to improve readability. (One is the rather odd, persistently wrong use of the word "any").

**Author's response :**

The minor comments below you pointed out about the English were corrected and the manuscript benefited as well from a deep proofreading from our co-author M. Theule, which is a native english speaker, which got rid of these problems and improved readability, as suggested.

**Author's changes in manuscript :**

See minor comments below ! The small changes operated throughout the manuscript were to many to get listed properly. Please see the new version of the manuscript.

Comments from Referees: The Abstract should summarize aims, study sites, methods, results and the most important conclusions. Currently, it starts with a weird sentence; some facts are missing (time period of investigation, basic information on the catchment); and the results are not well structured.

**Author's response :**

The abstract has been completed with missing information, and the key findings have been better underlined.

**Author's changes in manuscript :**

Basis information on the catchment being missing according to referee #2, the following sentence was added to the abstract : « ..., a small tributary valley with an active torrent system located exclusively in sedimentary rocks of the Chartreuse Massif (French Alps) ... ».

The text of the Abstract between L12 - 15 was modified such as it sticks well with the many changes that were made in the discussion and conclusion in order to be clear.

Comments from Referees: In the Study Area section, any information on climate is missing. Quite frequently throughout the paper the authors mention e.g. "response to heavy rainstorms that occur regularly" (P5 L5). What is a "heavy rainstorm" in the area? Information on mean annual precipitation, rainstorm intensities, MAAT is entirely missing. Further example P16 L9: "series of rainfall of low to moderate intensity" – it is never defined what low intensity is.

**Author's response :**

Information on mean annual precipitation (MAAT) will not provide much information for the analysis. That's why this was not explicitly mentioned. And high rainfall intensity or rainstorms refer to intense rainfalls from convective storms that commonly happen from June to September in all parts of the Alps.

But you're totally right, actually. These kinds of information have been thoroughly described in two PhD thesis (Theule, 2012 ; Loye, 2013), but never address in publication (for basis information, see Theule et al, 2012 and 2015, Loye et al. 2012 ; all cited in this paper). Therefore, this needs to be added.

**Author's changes in manuscript :**

The following figure here below was added at the place of the initial figure 3. This figure enables now to the reader to quantify what are series of rainfall of low to high intensity.

**Figure 3.** Maximum rainfall intensity over the monitoring period measured by a rain gauge located at the top of the torrent (calculated for a 5 minutes time interval). The mean annual precipitation is about 1500 mm in the headwater of the Manival (modified from Loye, 2013).

Comments from Referees: There is too much repetition and re-consideration of the own data in the Discussion towards the end, the paper gets tiresome to read. Consider to shorten the Discussion, with a stronger focus on the comparison to other work and on the geomorphic implications. More literature would be good in some places (e.g. P20 L14/15, and the entire sections 6.2 and 6.3). If other work is cited for comparison, it is mostly from French collegues working in the area. Most of paragraph 6.5 is rather speculative and could be left out (e.g. first sentences, and L11 ff)

**Author's response :**

Your suggestion (similar to the comments from other referees) was taking into account by reviewing the entire discussion. Repetition and some of the sentences summarizing the observed data were left out, as they were already mentioned in the results. The focus was put on clarifying the discussion (see our reply to general comments).

The many changes made in the discussion enable to get a comprehensive analysis and a clear picture of the seasonal variability of the debris supply and transfer that enables characterizing sediment dynamic and discussing the observation made.

Comparison to other works on the topic is not easy, since there is not so much published material about the seasonal fluctuations of sediment supply in alpine environments. Most literature related to the patterns and controls of debris supply, sediment dynamics and geomorphic processes in Alpine catchment prone to debris flows (like in the Illgraben, see the list below) were carried out either on longer observation period (several years or decades), providing a much larger pattern of hillslope and torrent activity, or at larger scale, providing either only the debris flow activity in the torrent or in the headwater or is only related to one particular slope instability (local active landslide for instance). Similar approaches that were carried out in geological, geomorphological and climatological environment pretty much different could not be considered as well, as these would make no sense, the scale of investigation being to precise and made in a too short period to be extrapolated or generalized. Still, we try to add some more references we thought they would illustrate the discussion.

Non exhaustive list of work that could have been considered :

- Imaizumi, F., Sidle, R.C., Tsuchiya, S., Ohsaka, O., 2006. Hydrogeomorphic processes in a steep debris flow initiation zone. Geophysical Research Letters 33.
- Cavalli, M., Trevisani, S., Comiti, F., Marchi, L., 2013. Geomorphometric assessment of spatial sediment connectivity in small Alpine catchments. Geomorphology 188, 31-41.
- Bennett, G.L., Molnar, P., McArdell, B.W., Schlunegger, F., Burlando, P., 2013. Patterns and controls of sediment production, transfer and yield in the Illgraben. Geomorphology 188, 68-82.
- Schürch, P., Densmore, A.L., Rosser, N.J., Lim, M., McArdell, B.W., 2011. Detection of surface change in complex topography using terrestrial laser scanning: application to the Illgraben debris-flow channel. Earth Surface Processes and Landforms 36, 1847-1859.

**Author's changes in manuscript :**

Chap 6.5 was entirely left out and only the key findings (even though one thinks this is « speculative ») of this chapter were mentioned in the other chapter of the discussion (see last § of the chap. 6.4)

Chap. 6.1 : literature was add at P19/L20

Chap. 6.2 : Literature was added at P21/L11)

Chap. 6.3 : Literature was add at P22/L16

Chap. 6.4 : Literature was add at P22/L27 and P23/L7 and by the end of this chapter.

Please, get to the chapter or paragraph related to the comment to have the whole picture of the many modifications made.

Comments from Referees: The Conclusion is the weakest section of the paper. Here, there would be the chance to summarize the multitude of details in a clear way, maybe using bullet points. However, the paragraph is as confusing as much of the Results and Discussion are. By the start of the Conclusions, the reader is desperately waiting for some clear, summarizing facts: Which are the key findings after such a lot of details? Furthermore, the Conclusions should summerize the already presented findings and NOT come up with new ideas (e.g. lines 5-7 are unclear and something new; wet snow avalanches (L16) have never been mentioned before, and so on). Quite a lot of the Conclusions (if not all) is in fact Discussion (e.g. P26 L1-4).

**Author's response :**

Your suggestion (similar to the comments from other referees) was taking into account by reviewing the entire conclusion. Elements that were in fact Disussion were suppressed to avoid repetition or put in the discussion subchapters related to such particular points, which should make the summarize of the already presented findings much more clear. As well, please refer to what was written in the author's general response to the most important comments (above).

**Author's changes in manuscript :**

See the new Conclusion chapter.

Comments from Referees: Table 1. TLS dates of acquisitions: In the text you speak of a "five-day survey" while here it seems to be one defined day for each survey. Do you mean "five surveys of one day each" in the text body? (If so, it seems like an impressive survey speed to achieve 50 scans from 9 viewpoints in a day..?)

Author's response : You're right, we were not able to achieve 50 scans from 9 viewpoints in a day !

Author's changes in manuscript : « survey date » become « starting dates of surveys » in the legend of Table 1.

Comments from Referees: Table 4. Sediment budget: It is not clear to me what the difference between "channel deposition input" and "sediment input" is; and what "total sediment" means

Author's response : « input » in Channel deposition input » is a mistake (bad location) and should be « channel deposition » and « total sediment input ». Now, this should be clear. In case, please refer to Theule et al. (2012) for more explanation.

Author's changes in manuscript: Table 4 , correction in the legend « Channel deposition » and Total sediment input »

Comments from Referees : Figure 1: The inset is not ideal as you can't see any topography at all, and a legend of the geological units is missing.

Author's response: The inset is here to show the geographic location of the catchment, the outset showing the topography. Information on the geology is given in details in Figure 2.

**Author's changes in manuscript : None !**

Comments from Referees : Figure 2: I assume that the thick blue lines are the main torrents. However, the white lines are not explained in the legend.

Author's response : see next comment.

Author's changes in manuscript : see next comment.

Comments from Referees: Figure 3: The Figure does not provide much important information and should be left out. I think that Fig. 1 and Fig. 4 are sufficient; maybe some of the names could be added in Fig. 1.

Author's response: We take your advise and decide to leave out Figure 3 by fusionning the information contained in Figure 3 into Figure 2 that displays the first-order debris flow channels (thick blue line) and their respective contributing area (white lines) as well. Hence, the names of the subcatchment are reported over Figure 2 too.

**Author's changes in manuscript :**

Modification of the Figure 2 as describes above. Modification of the legend as follow : "Geological map of the catchment headwater, after Gidon (1991) and location of first-order debris flow channels (thick blue line) of the Manival headwater (production zone of Fig. 1) and their respective contributing area (white lines)".

Comments from Referees : Fig. 4: Are the black symbols check dams?

Author's response : Yes, they are.

**Author's changes in manuscript : None !**

Comments from Referees : Fig. 7,8,9: It would be better to use the same colour charts throughout the three consecutive figures. Fig. 9 should use the same legend as in Fig. 7 and 8 (black circle around the rockslope erosion dots)

Author's response: Actually, all these figures do have the same legend in the original figures, but the bad rendering of the discussion paper make it different. This comes propably from the huge amount of points in Figure 8 in comparaison to figure 7 and 6. This will be checked out during the production process of the final version of the paper.

**Author's changes in manuscript : None !**

Comments from Referees: *Fig.* 10: It makes not much sense to show the data for each period - the entire study time (d) is sufficient.

Author's response : In the contrary, it makes sense to show that the observed debris production follows a power law distribution and that, even for short period of record, which has not been yet highlighted in the literature. So far, inventories known, like the one cited in the paper, were based on power law distribution taken from much longer inventories. But you bring out a good point. As this is not a central point of the paper and this is not discussed very much, we leave out plots a, b and c. This will contribute to reduce the size of the manuscript.

**Author's changes in manuscript :**

Plots a,b and c of Figure 10 are left out.

Comments from Referees : Figure 12: Very good figure - after some trying to understand, all important parts of the budget become clear.

Author's response : Thanks for the comment ...

Author's changes in manuscript : None !

Comments from Referees : Fig. 13,14,15 are excellent figures.

Author's response : Thanks again !

Author's changes in manuscript : None !

Comments from Referees : Fig. 16: leave out

Author's response : Why not ? This figure illustrates the inventory analysis using power law distribution of volume for potential rockfall (table 6) and hence the use of such inventory for prediction of debris supply and sediment processes, which is a key point of the paper. This is a willing of the authors to publish studies and works containing true application and original ways of characterizing surveys and records of sediment processes and geomorphic activity.

**Author's changes in manuscript : None !**

Comments from Referees : *Minor comments P5 L8: what does "any" mean in this context?* It means "no" Author's changes in manuscript : « any » become « no »

P5 L9: delete "a" -> okay !

*P5 L9: "old rock deposits flooring the hillslope side west (Fig. 3)" - This can't be seen in Fig. 3 (more in Fig. 4). I suggest to leave out Fig. 3.* Author's response : Already treated in previous comment

P5 L11: resulting in -> okay ! P5 L13: five (spell out numbers <13) -> okay !

P5 L26: rockfall CAN reach -> okay !

*P6 L17/18: "The entire coverage of the upper catchment" – later it becomes clear that the coverage is 'only' 84% of the deforested area. Thus, reword here.* **Author's changes in manuscript :** « entire » become « overall »

P7 L3: "procedure" instead of "manner" -> okay !

P7 L5: Each of the multiple scans -> okay !

P8 L1: computes -> okay !

P8 L4: what does "of similar sign" mean?

Author's response : The vector sign is defined in the L4 just above. It means : at least 8 adjacent points that have the same sign (either positive or negative = net change direction of topography, e.g. surface of erosion or deposition. We think this is clear enough ! Author's changes in manuscript : None !

*P9 L13: "cell differences lying below a given threshold" – which was this threshold?* Author's response : This threshold was the uncertainty of the measure, which is described in chapt. 3.4. Author's changes in manuscript : « threshold » become « uncertainty »

*P12 L19: "did not show any" instead of "showed any" ->* okay ! Author's changes in manuscript : « showed any » become « did not show any »

P13 L15: spatial extent -> okay ! P13 L18: did not show any -> okay ! Author's changes in manuscript : « any » become « no »

P13 L22: signs -> okay ! P13 L23: rilling areas -> okay ! P14 L10: rock slope erosion -> okay ! P14 L20: rockfalls -> okay ! P15 L13: "almost without any" instead of " with almost any" -> okay ! P16 L5: in the torrent -> okay ! P16 L9: "no" instead of "any" -> okay ! P16 L14: rainfalls -> okay ! P16 L19: "no" instead of "any" -> okay ! P16 L20: "No" instead of "Any"-> okay ! P17 L1: Synthesis -> okay ! P17 L8: "of which" instead of "whereas" -> okay ! P17 L14 "acts as recharging the torrent" - reword, what does this mean? Author's response : This was had english to say « contributed to recharge

Author's response : This was bad english to say « contributed to recharge the torrent with sediment » Author's changes in manuscript : sentence changed to « During the autumn, bedload transport of several hundreds of m3 contributed to recharge the torrent with sediment all along. »

P18 L15: delete "degradation" -> okay !

P18 L21: "the bedrock surface is often highly fractured, suggesting frost shattering."This is a case of wrong reasoning and self-fulfilling prophecy (which is frequently found in this context). The fact that the bedrock surface is highly fractured is, it itself, no indication of frost action. It might be fractured for tectonic-lithological reasons, or by the impact of any other type of weathering (hydration, solution processes, wetting/drying...)

Author's response : You're right, that why the next sentence says : « The spatial pattern of rockfall strongly suggests also a lithologic influence that can be explained by differential erosion between the successive limestone and marl beds. In the rock walls series side west, the monoclinal configuration of the bedding, combined with a strong difference of competency between stratigraphic sequences, give rise to overhanging formation highly susceptible to failure ». So, this must be a bit of all the processes you describe.

Author's changes in manuscript : « lithologic » become « tectonic-lithological »

P20 L3: "No" instead of "Any" -> okay !

P20 L27: "For the entire area" instead of "globally" (which would be earth-wide) -> You're right ! P21 L21: "sediment density" instead of "process density"? -> okay ! P23 L7: exhaustion instead of starvation -> okay ! P23 L8: "no" instead of "any" -> okay !

**Anonymous Referee #3 Received and published: 23 February 2016**

Thank you very much for your comments and relevant suggestions. Please find below our answers.

Comments from Referees : The paper would gain being presented in a more straightforward way, and would benefit of an English editing

**Author's response :**

Several aspects of the manuscript were already considered according to the previous comments of the two first referees. As well, many mistakes regarding the language were corrected according to the comments made by the second referee.

**Author's changes in manuscript :**

An English revision was done by a native English speaker.

Comments from Referees: P. 2, line 25: could you specify how you consider snowmelt identical to rainfall? Snowmelt has been proven as a different triggering factor from rainfall (intense or long lasting) (ex. Decaulne et al., 2005 in Geografiska Annaler), and has also been denied, or with a very secondary action in some studies (ex Jomelli 2004 in Climate Change).

**Author's response :**

incl. snowmelt means simply that in mountain catchment such as in the Manival, runoff is very often coupled to snowmelt in debris-flow channels during some parts of the year. That's it ! But whatever the literature say (the paper of Decaulne et al, 2005 is in a complete other climatic context), I know a debris flow channel near my town in Wallis, Swizerland, where snowmelt had been of great influence in some debris flow events (google : E. Bardou, torrent de Tracuit, Zinal, lave torrentielle).

In one of your comment later (the one about P. 18, line 18), you notice "You may also consider the triggering factors for debris flow initiation, and especially snowmelt of snow avalanche deposits (Bardou and Delaloye, 2004 in NHESS)". So we are not sure we get your comment properly.

**Author's changes in manuscript :**

We can leave out « (incl. snowmelt) » whether this is inconsistent with the literature.

Comments from Referees : *P. 3, line 3-4: how do you compare debris flow amounts with initiation mechanisms in a quantitative way?*

**Author's response :**

Please be more specific ? Whatever the mechanisms of initiation, an important aspect of debris flow is the amount of debris ; nothing really quantitative to compare behind this ...

**Author's changes in manuscript : None !**

Comments from Referees : *P. 6, line 27: what are the criteria to recognize and then eliminate erroneous points?*

**Author's response :**

No specific criteria, but you can just see it in the 3D point cloud image according to the shape and intensity of the image data. This is pretty easy to recognize and when you are not sure, you can drap a photograph over the 3D point cloud, which give you an image with real surface color, almost like if you were on the field. Erroneous points were removed manually (as notice in P7 L1)

**Author's changes in manuscript : None**

Comments from Referees : *P. 6, line 17-20: has this be previously validated by experiments? Please use a reference. P. 10: equations should be numbered.*

**Author's response :**

Yes, the manufacturer (optech Inc.) has made his own experiment. And our lab made our own experiments as well. See :

- Abellán, A., Jaboyedoff, M., Oppikofer, T., Vilaplana, J.M., (2009). Detection of millimetric deformation using a terrestrial laser scanner: experiment and application to a rockfall event. Natural Hazards & Earth System Science 9, 365–372.

- Jaboyedoff, M., Oppikofer, T., Abellán, A., Derron, M.-H., Loye, A., Metzger, R., and Pedrazzini, A. (2012). Use of LIDAR in landslide investigations: a review. Natural Hazards, p. 1-24. DOI 10.1007/s11069-010-9634-2.

Equations were numbered as required by the journal in the text file that I sent, but apparently this is not showed in the format of the "printer-frendly Version".

**Comments from Referees :** *P. 12, line 4 'whereas b tends to be rather site independent'; why don't you say 'is site independent'? why are you so cautious?*

**Author's response :**

Because people that have worked on this aspect of the equation came to the conclusion that b tends to be rather site independent. But enough case study are still missing for tough comparison.

Comments from Referees : *P. 12, line 16: what make you select the duration of each period, as they are uneven in time. Please be more specific on this and provide an explanation earlier in the pape.*

**Author's response :**

Sorry, but we don't understand the comment as what you say has nothing to do with what is written on P12/L16.

Comments from Referees: *P. 18, line 18: you specify that freeze-thaw is a major key for debris supply, contributing to rockfall that will later feed the debris flows. You may also consider the triggering factors for debris flow initiation, and especially snowmelt of snow avalanche deposits (Bardou and Delaloye, 2004 in NHESS).*

**Author's response :**

Your comment make sense, but the sentence you comment is in a chapter that talks about debris production. Triggering factors for debris flow initiation are discussed later in the discussion chapters.

**Comments from Referees :** *P. 20, line 8: you mention 'about' before the calculation; that's a quite relevant word, which might lead to very different result according to the mentionned quantities.*

Author's response: You're right ! a difference in the rate of rockfall erosion of a few mm yr-1 make a difference in volume in sediment supply. About means « approximatively » in this context.

Author's changes in manuscript : « About » becomes « approximatively »

Lausanne, March 25, 2016

On behalf of all the authors

Dr. Alexandre Loye

---

## Author Response (AR2)

**Reply to final review on "Headwater sediment dynamics in debris flow catchment: implication of debris supply using high resolution topographic surveys" by A. Loye et al.**

Submitted to Earth Surf. Dynam. (2015-48, 2016), Special Issue: Frontiers in geomorphometry.

**Response to the final review**

Thank you very much for the comments and particularly for the very nice review of the English made by handling associate Editor Ms Susan Conway that improved the readability and clarity of the manuscript. Please find here below a point-by-point reply to the comments, the suggestions and the few minor corrections addressed by her review.

**P1/L24**: As a willing to keep showing to the reader the contradictory of the behaviour in sediment transfer, the word « equivocal » was changed to « ambiguous » in order to keep the idea « open to discussion » or « to interrogate ».

**P2/L6 :** Rainfall, snowmelt, intensity, duration, each of these caracteristics of precipitation plays a role by itself, but remains unsufficient to predict debris flow. That's why « intensity » and « duration » are emphasized. The word « precipitation » would be too general. No change was made !

**P2/L9-10**: The sentence was replaced by the second proposition of the reviewer, which reach the closest of the authors'idea.

**P2/L19**: Sediment process activity means all what the reviewer suggested : process rates, frequency, volumes, etc. Therefore, the term was replaced with « sediment process and rates and volumes » in order to emphasize the idea « activity ».

**P2/L26-27**: The confusing sentence was removed as suggested.

P3/L11-12 : The « poor-english » sentence was modified as suggested.

**P4/L17**: No, this map was made by the first author and has not been published yet (beside in his unpublished PhD manuscript).

**P5/L6-7**: Yes, this is what we mean. The add was made as suggested.

**P5/L19**: Yes, this is what we mean. The add was made as suggested.

**P5/L28**: The proposition of the reviewer is correct. The change was made as suggested.

**P6/L4**: The standard national grid for France was used. No need to specify the coordinate system in full.

**P7/L23-24**: Both difference (positive and negative) were included in the sum together. This was specified as suggested.

**P7/L24**: A reference to the later section defining the treshold was added as suggested.

**P8/L12**: The correct word is « consistent ». The change was made as suggested.

**P9/L13**: The suggestion is accurate. It makes sense to show a clear difference. The modification was kept.

P10/L25: It was meant that the gravel-wedges are incised and the related material was entrained and re-deposited further down. The « poor-english » sentence was rephrased by simply combining the two last sentences, that should emphasized the process observed in the field more clearly for the reader. Of course, this sentence could be removed, but the first author thinks that this is important to keep the reader informed of the several main observations made in the field.

**P12/L30-31**: Yes, this is what we mean, but there are several classes of volume, not just the largest one. A reference to the table 6 was added for clarity.

**P13/L18 and L21-23 :** By competent, we mean : « capable of transfering sediment downward, but with a relevant amount of sediment, not just a very small transfer of sediment that occurs every time. The word « competent » was replaced by the word « significant » and the word « downstream » was added to show the difference between the up (distal) and down (proximal) part of the torrent.

**P17/L12**: The reviewer is right, the two concepts are kind of the same. The sentence was rephrased in two distinct sentences to make the meaning clearer. The word « whereas » was written to give an idea of contrast, but there is not much contrast to express. A « and » make more sense.

**P17/L14**: (same as P13/L18). Here the word « competent » was replaced by « effective » because of the idea of the effectivity and efficiency of sediment transfer.

**P19/L6 :** ... from the head ? « of the subcatchment ». This was added to the sentence.

**P20/L8 :** subordinated means in this context « made dependent/conditional ». The word « subordinated » was replaced by the word « made conditional on ».

**P20/L9-10**: The term « early in the season » was replaced by « early in the year, from winter to spring » and the term « late in the season » was replaced by « later in the year », as suggested to make the temporality more clear for the reader.

Other comments: All minor corrections and changes according to poor english, bad synthaxes and small orthographic mistakes suggested by the associate Editor Ms Susan Conway was integrated to the text of the manuscript without any exception.

**Table 2 :** The last column is the total surface area of the non-vegetated area in [%] that was covered by the Lidar data survey. This means, for instance in the first row, that the scanned area covers the 84% of the surface of the subcatchment uncovered with vegetation. This is not the same as the third column,

which informs about the vegetation cover of the subcatchment. For clarity, the header of the last column was replaced by « [%] of the non-vegetated surface ».

**Table 3 :** The comment arises probably because the sentence says not exactly what the authors want to say due to bad english synthax. The authors wanted to say : 1. ... to detect distinctively geomorphic features with topographic changes down to a certain threshold/limit ; 2. By consequence, geomorphic features with a given minimum volume can be detected.

Therefore, the correct sentence that was placed instead, is : « subsequently to detect topographic changes down to a certain minimum volume of geomorphic features ».

More generally to the comment, the authors are convinced that this part was sufficiently detailed in the manuscript. The whole section 3.4 is dedicated to that, which is illustrated by table 3 and figure 6. More precisely, L.18 and L.27 on page 8 treat of this subject and the following lines of L.27 talk about the ability of the process to create a distinct topographic change. But we are aware that point cloud accuracy and limits of detection are two concepts that are not easy to understand for the ones who have never co-registered and co-georeferenced Lidar scanning data.

**Figure 1 :** The inset is a map of situation showing where the catchment is located. All useful information are in : it shows the country, the region, the names of the different city in the vicinity and it shows the different mountain ranges and it shows that the catchment is located in the prealpine domain at the border with the cristalline belt (also, the Alps). A simple contour map would not have not brought up that much useful informations to the people who don't know the area. The inset was kept unchanged !

The source of the aerial image and DEM were credited.

Figure 3 : As suggested, the location of the rain gauge was added, but in figure 4.

**Figure 5 :** Some kind of scale, as suggested, was added to part A of the figure. As the part B is a illustration and is schematic, the scale is actually not defined as it can be any scales. So adding something about scale is not really necessary to our opinion. No information about the scale were added !

Figure 7,8,9 : LoD was added in the legends, as suggested, instead of zero.

**Other comments :** All minor corrections and changes according to poor english, bad synthaxes and small orthographic mistakes that were suggested by the associate Editor Ms Susan Conway in caption and text of the tables and figures were integrated without any exception.

Lausanne, Mai 19, 2016

On behalf of all the authors

Dr. Alexandre Loye

---

## Author Response (AR3)

**Reply to editor comments on "Headwater sediment dynamics in debris flow catchment: implication of debris supply using high resolution topographic surveys" by A. Loye et al.**

Submitted to Earth Surf. Dynam. (2015-48, 2016), Special Issue: Frontiers in geomorphometry.

**Response to the last comments before publication**

The comment about the title was considered as suggested, as we think to be a very good suggestion.

The reviewer's efforts were acknowledged as it should be in the chapter dedicated to that at the end of the manuscript.

The correction was made p. 20/L.14 as asked.

Lausanne, June 04, 2016

On behalf of all the authors

Dr. Alexandre Loye